# Make Some Noise: Reliable and Efficient Single-Step Adversarial Training

**Pau de Jorge** *
University of Oxford
NAVER LABS Europe

**Adel Bibi**
University of Oxford

**Riccardo Volpi**
NAVER LABS Europe

**Amartya Sanyal**
ETH Zürich
ETH AI Center

**Philip H. S. Torr**
University of Oxford

**Grégory Rogez**
NAVER LABS Europe

**Puneet K. Dokania**
University of Oxford
Five AI Ltd.

## Abstract

Recently, Wong et al. [35] showed that adversarial training with single-step FGSM leads to a characteristic failure mode named *Catastrophic Overfitting* (CO), in which a model becomes suddenly vulnerable to multi-step attacks. Experimentally they showed that simply adding a random perturbation prior to FGSM (RS-FGSM) could prevent CO. However, Andriushchenko and Flammarion [1] observed that RS-FGSM still leads to CO for larger perturbations, and proposed a computationally expensive regularizer (GradAlign) to avoid it. In this work, we methodically revisit the role of noise and clipping in single-step adversarial training. Contrary to previous intuitions, we find that using a *stronger noise* around the clean sample combined with *not clipping* is highly effective in avoiding CO for large perturbation radii. We then propose *Noise-FGSM* (N-FGSM) that, while providing the benefits of single-step adversarial training, does not suffer from CO. Empirical analyses on a large suite of experiments show that N-FGSM is able to match or surpass the performance of previous state-of-the-art GradAlign, while achieving $3\times$ speed-up. Code can be found in `https://github.com/pdejorge/N-FGSM`

## 1 Introduction

Deep neural networks have achieved remarkable performance on a variety of tasks [12, 27, 7]. However, it is well known that they are vulnerable to small worst-case perturbations around the input data – commonly referred to as *adversarial examples* [31]. The existence of such adversarial examples poses a security threat to deploying models in sensitive environments [2, 4]. This has motivated a large body of work towards improving the *adversarial robustness* of neural networks [11, 22, 32, 25, 5].

The most popular family of methods for learning robust neural networks is based on the concept of *adversarial training* [11, 20]. In a nutshell, adversarial training can be posed as a min-max problem where instead of minimizing some loss over a dataset of *clean* samples, we augment the inputs with worst-case perturbations that are generated online during training. However, obtaining such perturbations is NP-hard [34] and hence, different *adversarial attacks* have been suggested that approximate them. In their seminal work, Goodfellow et al. [11] proposed the *Fast Gradient Sign Method* (FGSM), that generates adversarial attacks by performing a gradient ascent step on the loss function. Yet, while FGSM-based adversarial training provides robustness against single-step FGSM adversaries, Tramèr et al. [32] showed that these models are still vulnerable to multi-step attacks, namely those allowed to perform multiple gradient ascent steps. Given their better (robust)

---

*Correspondence to `pau@robots.ox.ac.uk`

36th Conference on Neural Information Processing Systems (NeurIPS 2022).

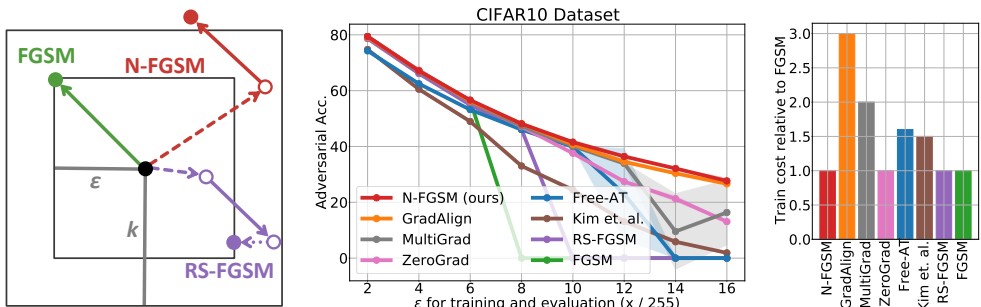

Figure 1: **Left:** Visualization of FGSM [11], RS-FGSM [35] and N-FGSM (ours) attacks. While RS-FGSM is limited to noise in the $\epsilon - l_\infty$ ball, N-FGSM draws noise from an arbitrary $k - l_\infty$ ball. Moreover, N-FGSM does not clip the perturbation around the clean sample. **Middle:** Comparison of single-step methods on CIFAR-10 with PreactResNet18 over different perturbation radii ($\epsilon$ is divided by 255). Our method, N-FGSM, can match or surpass state-of-the-art results while *reducing the cost by a* $3\times$ *factor*. Adversarial accuracy is based on PGD-50-10 and experiments are averaged over 3 seeds. **Right**: Comparison of training costs relative to FGSM baseline based on the number of Forward-Backward passes, see Appendix Q for details.

performance, multi-step attacks such as *Projected Gradient Descent* (PGD) [20] have now become the de facto standard for adversarial training.

The main downside of multi-step adversarial training is that the cost of these attacks increases linearly with the number of steps, making their applicability often computationally prohibitive. For this reason, several works have focused on reducing the cost of adversarial training by approximating the worst-case perturbations with single-step attacks [35, 26, 33]. In particular, Wong et al. [35] studied FGSM adversarial training and discovered that it suffers from a characteristic failure mode, in which a model suddenly becomes vulnerable to multi-step attacks despite remaining robust to single-step attacks. This phenomenon is referred to as *Catastrophic Overfitting* (CO). As a solution, they argued that adding a random perturbation prior to FGSM (RS-FGSM) seemed sufficient to prevent CO and produce robust models. Yet, Andriushchenko and Flammarion [1] recently observed that RS-FGSM still leads to CO as one increases the perturbation radii of the attacks. They suggested a regularizer (GradAlign) that can avoid CO in the settings they considered, but at the expense of computing a double derivative – significantly increasing the computational cost with respect to RS-FGSM.

In this paper, we revisit the idea of including noise in single-step attacks. Differently from previous methods that consider the noise as part of the attack, we propose an adversarial training procedure where the noise is used as a form of *data augmentation*. As we detail in Section 4, this motivates us to introduce two main changes with respect to previous methods: 1) We center adversarial perturbations with respect to noise-augmented samples and therefore, unlike previous RS-FGSM, we do not clip around the clean samples. 2) We use noise perturbations larger than the $\epsilon-$ball, since they are not restricted by the strength of the attack anymore. Our experiments show that performing data augmentation with sufficiently *strong noise* and removing the *clipping step* improves model robustness and prevents CO, even against large perturbation radii. Our new method, termed N-FGSM, matches, or even surpasses, the robust accuracy of the regularized FGSM introduced by Andriushchenko and Flammarion [1] (GradAlign), while *providing a* $3\times$ *speed-up*.

To corroborate the effectiveness of our solution, we present an experimental survey of recently proposed single-step attacks and empirically demonstrate that N-FGSM trades-off robustness and computational cost better than any other single-step approach, evaluated over a large spectrum of perturbation radii (see Figure 1, middle and right panels), over several datasets (CIFAR-10, CIFAR-100, and SVHN) and architectures (PreActResNet18 and WideResNet28-10). We will release our code to reproduce the experiments.

## 2 Related Work

Since the discovery of adversarial examples, many defense mechanisms have been proposed, *adversarial training* being one of the most popular and empirically validated. We can categorise adversarial

training methods based on how they approximate the perturbations applied to training samples. *Multi-step* approaches approximate an inner maximization problem to find the worst-case perturbation with several gradient ascent steps [37, 18, 20]. While this provides a better approximation, it is also more expensive. At the other end of the spectrum, *single-step* methods only use one gradient step to approximate the worst case perturbation. Goodfellow et al. [11] first proposed FGSM; Tramèr et al. [32] proposed a new variant with an additional random step (R+FGSM), but observed that both methods were vulnerable to multi-step attacks. Shafahi et al. [26] proposed *Free Adversarial Training* (Free-AT), which successfully reduced the computational cost of training by using a single backward pass to compute both weight update and attack. Motivated by this, Wong et al. [35] explored a variant of R+FGSM, namely RS-FGSM, that uses a less restrictive form of noise and showed this can improve robustness for moderate perturbation radii at the same cost as FGSM. Recently, Andriushchenko and Flammarion [1] proposed the GradAlign regularizer. Combining FGSM with GradAlign results in robust models at even larger perturbation radii. However, GradAlign suffers from a *threefold* increase in the training cost to as compared to FGSM. The need for more efficient solutions has motivated a growing body of work whose goal is the design of computationally lighter single-step methods [10, 15, 33, 23, 19].

In this work, we revisit the idea of combining noise with the FGSM attack. Our method builds upon FGSM and intuitions from R+FGSM and RS-FGSM to combine it with random perturbations, however, we consider the noise step as data augmentation rather than part of the attack. This motivates us to use a stronger noise *without* clipping. As opposed to [14], our thorough study leads to a practically effective approach that yields robustness also against large perturbation radii.

## 3 Preliminaries on Single-Step Adversarial Training

Given a classifier $f_\theta : \mathcal{X} \to \mathcal{Y}$ parameterized by $\theta$ and a perturbation set $\mathcal{S}$, $f_\theta$ is defined as *robust* at $x \in \mathcal{X}$ on the set $\mathcal{S}$ if for all $\delta \in \mathcal{S}$ we have $f_\theta(x + \delta) = f_\theta(x)$. One of the most popular definitions for $\mathcal{S}$ is the $\epsilon - \ell_\infty$ ball, *i.e.,* $\mathcal{S} = \{\delta : \|\delta\|_\infty \leq \epsilon\}$. This is known as the $l_\infty$ threat model which we adopt throughout this work.

To train networks that are robust against $\ell_\infty$ threat models, adversarial training modifies the classical training procedure of minimizing a loss function over a dataset $\mathcal{D} = \{(x_i, y_i)\}_{i=1:N}$ of images $x_i \in \mathcal{X}$ and labels $y_i \in \mathcal{Y}$. In particular, adversarial training instead minimizes the worst-case loss over the perturbation set $\mathcal{S}$, *i.e.,* training is on the adversarially perturbed samples $\{(x_i + \delta_i, y_i)\}_{i=1:N}$. Under the $l_\infty$ threat model, we can formalize adversarial training as solving the following problem:

$$\min_\theta \sum_{i=1}^N \max_\delta \mathcal{L}(f_\theta(x_i + \delta), y_i) \text{ s.t. } \|\delta\|_\infty \leq \epsilon, \tag{1}$$

where $\mathcal{L}$ is typically the cross-entropy loss. Due to the difficulty of finding the exact inner maximizer, the most common procedure for adversarial training is to approximate the worst-case perturbation through several PGD steps [20]. While PGD has been shown to yield robust models, its cost increases linearly with the number of steps. As a result, several works have focused on reducing the cost of adversarial training by approximating the inner maximization with a single-step.

If the loss function is linear with respect to the input, the inner maximization of Equation (1) will enjoy a closed form solution. Goodfellow et al. [11] leveraged this to propose FGSM, where the adversarial perturbation follows the direction of the sign of the gradient. Tramèr et al. [32] proposed adding a random initialization prior to FGSM. However, both methods were later shown to be vulnerable against multi-step attacks, such as PGD. Contrary to prior intuition, recent work from Wong et al. [35] observed that combining a random step with FGSM can actually lead to a promising robustness performance. In particular, most recent single-step methods approximate the worst-case perturbation solving the inner maximization problem in Equation (1) with the following general form:

$$\delta = \psi\Big(\eta + \alpha \cdot \text{sign}\big(\nabla_{x_i}\mathcal{L}(f_\theta(x_i + \eta), y_i)\big)\Big), \tag{2}$$

where $\eta$ is drawn from a distribution $\Omega$. For example, when $\psi$ is the projection operator onto the $\ell_\infty$ ball and $\Omega$ is the uniform distribution $[-\epsilon, \epsilon]^d$, where $d$ is the dimension of $\mathcal{X}$, this recovers RS-FGSM. Under a different noise setting where $\Omega = (\epsilon - \alpha) \cdot \text{sign}(\mathcal{N}(\mathbf{0}_d, \mathbf{I}_d))$ and by choosing the step size $\alpha$ to be in $[0, \epsilon]$, we recover R+FGSM by Tramèr et al. [32]. This was among the first works to explore the application of noise to FGSM, but did not report improvements over it. If we consider $\Omega$ to be deterministically 0 and $\psi$ to be the identity map, we recover FGSM. Finally, if we take FGSM and add a gradient alignment regularizer, this recovers GradAlign.

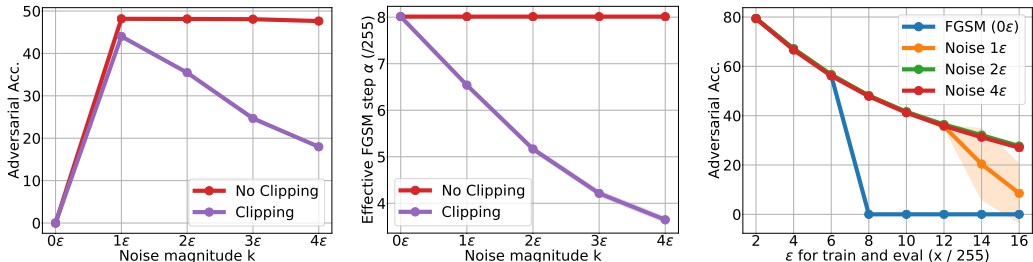

Figure 2: **Left:** Ablation of clipping vs not clipping around the clean sample $x$ for $\epsilon = {}^8/_{255}$. Clipping leads to a significant drop in robustness which increases with the strength of the noise augmentations. **Middle:** Analysis of the effective FGSM step size after clipping. We observe that clipping leads to a decrease in the effective FGSM step size, thus, adversarial perturbations will be more similar to random noise. **Right:** N-FGSM (ours) when varying the noise magnitude $k$ ($\epsilon$ is divided by 255). Increasing the amount of noise is key to avoiding CO. For (left) and (right) plots, adversarial accuracy is based on PGD-50-10 and experiments are averaged over 3 seeds.

## 4 Noise and FGSM

Previous methods that combined noise with FGSM, *e.g.,* R+FGSM [32] and RS-FGSM [35], have considered the noise as a *random step* integrated within the attack. Since it is a common practice to restrict adversarial perturbations to the $\epsilon-$ball, we argue that this introduces a trade-off between the magnitude of the random step and that of the attack. For illustration, consider the purple lines corresponding to RS-FGSM in Figure 1 (left). If the initial random step is significantly larger than the $\epsilon-$ball, then the final clipping step will have a noticeable impact on the perturbation, possibly removing a considerable portion of the FGSM step (middle arrow). To prevent this from happening, R+FGSM and RS-FGSM restrict the random step to lie within the $\epsilon-$ball, thereof implicitly entangling the noise magnitude and the attack strength.

Contrary to previous methods, we note that adding noise to the clean sample can be considered as a form of *data augmentation* to be applied independently from the attack. We make two considerations from this perspective 1) When one performs data augmentation during adversarial training, the input after the corresponding transformation is the starting point to compute the adversarial perturbations, therefore, we argue that adversarial attacks should be centered around the noise-augmented samples. This motivates us to *avoid clipping* around the clean sample. 2) We do not need to restrict the noise augmentation to lie inside an $\epsilon-$ball, since its strength is disentangled from that of the attack. Thus, we can use *stronger noise-augmentations* than previous methods.

These modifications lead to a novel adversarial training method that combines noise-based data augmentations with FGSM. We denote it as Noise-FGSM (N-FGSM). Following the notation introduced in Section 3, we define the noise augmented sample as $x_{\text{aug}} = x + \eta$ where $\eta$ is sampled from a uniform distribution on $[-k, k]^d$ (where we can have $k > \epsilon$). Then the adversarially perturbed samples have the following form:

$$x_{\text{N-FGSM}} = x_{\text{aug}} + \alpha \cdot \text{sign}\big(\nabla_{x_{\text{aug}}} \mathcal{L}(f_\theta(x_{\text{aug}}), y)\big). \tag{3}$$

This construction corresponds to augmenting the clean sample $x$ with the perturbation defined in Equation (2) where $\psi$ is the identity map and $\Omega$ is the uniform ditribution spanning $[-k, k]^d$. We detail our full adversarial training procedure in Algorithm 1. In what follows, we analyse the effect of treating the noise as data augmentation as opposed to treating it as a random step within the attack. In particular, we show that clipping around the clean sample $x$ (as done in RS-FGSM) can strongly degrade the robustness of the network. Moreover, we show that as we increase the $\epsilon$ radii of adversarial attacks, we need stronger noise perturbations than previously used to prevent CO.

**Clipping around clean sample $x$ hinders the effectiveness of perturbations.** We analyse two variants, one where perturbations are clipped around the clean sample $x$ (as done in previous methods) and another where no clipping is applied. In Figure 2 (left), we report the robust accuracy using PGD-50-10 (*i.e.,* PGD attack with 50 iterations and 10 restarts) with $\epsilon = {}^8/_{255}$ and observe that clipping significantly degrades the effectiveness of FGSM training. To understand this drop, consider the following perturbations; (**1**) a baseline perturbation where we only use noise $\delta_{\text{random}} = \psi(\eta)$ and (**2**) a per-

---

**Algorithm 1** N-FGSM adversarial training

---

1: **Inputs:** epochs $T$, batches $M$, radius $\epsilon$, step-size $\alpha$ (default: $\epsilon$), noise magnitude $k$ (default: $2\epsilon$).
2: **for** $t = 1, \ldots, T$ **do**
3:    **for** $i = 1, \ldots, M$ **do**
4:       $\eta \sim \text{Uniform}[-k, k]^d$
5:       $x_{\text{aug}}^i = x^i + \eta$ // *Augment sample with additive noise.*
6:       $x_{\text{N-FGSM}}^i = x_{\text{aug}}^i + \alpha \cdot \text{sign}\big(\nabla_{x_{\text{aug}}^i} \mathcal{L}(f_\theta(x_{\text{aug}}^i), y^i)\big)$ // *N-FGSM augmented sample.*
7:       $\nabla_\theta = \nabla_\theta \mathcal{L}(f_\theta(x_{\text{N-FGSM}}^i), y^i)$ // *Compute gradients of model's weights*
8:       $\theta = \text{optimizer}(\theta, \nabla_\theta)$ // *Standard weight update, (e.g., SGD)*

---

turbation that combines noise with FGSM $\delta_{\text{full}} = \psi(\eta + \alpha \cdot \text{sign}\big(\nabla_x \mathcal{L}(f_\theta(x + \eta), y)\big))$. Moreover, we consider two cases in which we either define $\psi$ as a clipping operator or as the identity. We define the effective FGSM step size as the magnitude corresponding to the ratio[2] $\alpha_{\text{effective}} = \|\delta_{\text{full}} - \delta_{\text{random}}\|_2 / \|x\|_2$ which measures the contribution of the FGSM step in the final perturbation compared to simply following the noise direction $\eta$. In Figure 2 (middle), we observe that the clipping operator reduces the effective magnitude of FGSM, thus, perturbations become more similar to only using random noise. On the other hand, without clipping we always take the full step in the FGSM direction. This highlights the trade-off between noise magnitude and attack strength discussed above.

**Larger noise is also necessary to prevent CO.** As discussed above, previous work did not investigate the effects of using noise perturbations potentially larger than the attack strength. However, we empirically find that increasing the noise magnitude is key to avoiding CO. In particular, as seen in Figure 2 (right), when no clipping is performed, it is crucial that we augment with larger noise magnitude in order to prevent CO in all settings. We find the noise magnitude of $k = 2\epsilon$ to work well in most of our experiments, however, a more extensive hyperparameter tuning might improve our results further.

Note that these results are contrary to previous intuitions: Andriushchenko and Flammarion [1] suggested that the random step in RS-FGSM is not important per se, arguing that its main role is reducing the $\ell_2$ norm of the perturbations, so that the loss remains to be approximately locally linear. In contrast, N-FGSM perturbations are larger on expectation than those of RS-FGSM, while they do not suffer from CO (refer Appendix N). We believe that our findings will lead to a better understanding of the role of noise in avoiding CO in future work. Moreover, in Section 6 we conduct extensive analyses to show that, despite N-FGSM obtains larger perturbations, clean accuracy does not degrade and other methods do not benefit from simply increasing the strength of their attacks.

**Why does noise augmentation avoid CO?** Andriushchenko and Flammarion [1] found that after CO, the gradients of the loss with respect to the input around clean samples became strongly misaligned, which is a sign of non-linearity. Moreover, Kim et al. [15] showed that the loss surface of models suffering from CO appears distorted, *i.e.,* there is a sharp peak in the loss surface along the FGSM direction, which seems to render FGSM ineffective (observe from Figure 5 how after after CO, visually, FGSM perturbations change drastically). In order to prevent CO, GradAlign explicitly regularizes the loss surface so it remains linear. To investigate further, we plot the loss surface at the end of training for different methods (see Figure 14 in Appendix) and find that, while FGSM or RS-FGSM lead to a distorted loss, N-FGSM obtains a non-distorted loss surface similar to that obtained by GradAlign regularizer. Thus, it seems that adding strong noise-augmentations implicitly regularizes the loss landscape, leading to more effective single-step attacks. This aligns with previous work that theoretically link noise augmentations with a regularizer that encourages Lipschitzness [3].

## 5 Robustness Evaluations and Comparisons

We compare N-FGSM against several adversarial training methods, on a broad range of $\epsilon - l_\infty$ radii. Following Wong et al. [35], we evaluate adversarial robustness on CIFAR-10/100 [16] and SVHN [21] with PGD-50-10 attacks, using both PreactResNet18 [13] and WideResNet28-10 [36]. Evaluations with AutoAttack[6] are also in Appendix G.

---

[2]The denominator $\|x\|_2$ is simply to normalize the $\ell_2-$norm and be comparable to the FGSM step size $\alpha$.

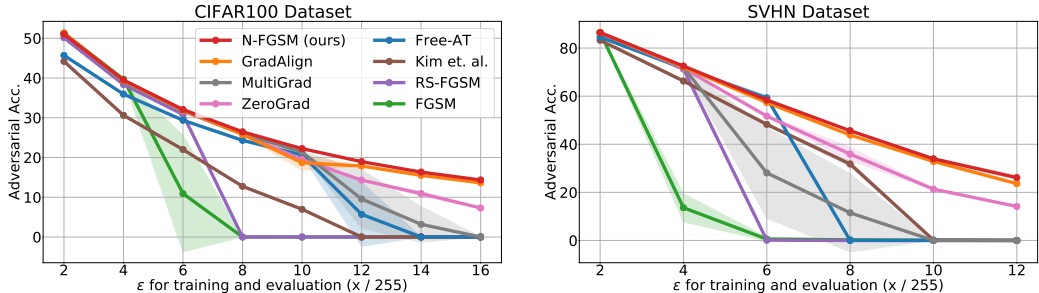

Figure 3: Comparison of single-step methods on CIFAR-100 (left) and SVHN (right) with PreactResNet18 over different perturbation radius ($\epsilon$ is divided by 255). Our method, N-FGSM, can match or surpass prior art results while *reducing the cost by a $3\times$ factor*. Adversarial accuracy is based on PGD-50-10 and experiments are averaged over 3 seeds. Legend is shared among plots.

## 5.1 Comparison against Single-Step Methods

We start by comparing N-FGSM against other single-step methods. Note that not all single-step methods are equally expensive, since they may involve more or less computationally demanding operations. For instance, GradAlign uses a regularizer that is considerably expensive, while MultiGrad requires evaluating input gradients on multiple random points. For a comparison of training costs of different single-step methods, we refer the reader to Figure 1 (right). We use RS-FGSM and Free-AT with the settings recommended by Wong et al. [35]. We apply GradAlign with hyperparameters reported in the official repository[3]. ZeroGrad and Kim et al. [15] do not have a recommended set of hyperparameters; for a fair comparison we ablate them and select the ones with highest adversarial accuracy (for every $\epsilon$ and dataset). We train on CIFAR-10/100 for 30 epochs and on SVHN for 15 epochs with a cyclic learning rate. Only for Free-AT, we use 96 and 48 epochs for CIFAR-10/100 and SVHN, respectively, to obtain comparable results following Wong et al. [35]. CIFAR-10 results are in Figure 1 (middle), whereas CIFAR-100 and SVHN are in Figure 3.

As observed in Figure 1 and Figure 3, FGSM and RS-FGSM suffer from CO for larger $\epsilon$ attacks on all reported datasets. For instance, RS-FGSM fails against attacks with $\epsilon = {}^8\!/{}_{255}$ on CIFAR-10 and CIFAR-100 and against $\epsilon = {}^6\!/{}_{255}$ on SVHN. With appropriate hyperparameters, ZeroGrad is able to consistently avoid CO. However, it obtains sub-par robustness compared to N-FGSM and GradAlign, especially against large $\epsilon$ attacks. Neither MultiGrad nor Kim et al. [15] avoid CO in all settings despite being more expensive. Free-AT also suffers from CO on all three datasets as also observed by Andriushchenko and Flammarion [1]. In contrast, N-FGSM avoids CO on all datasets, achieving comparable or superior robustness to GradAlign *while being 3 times faster*.

## 5.2 Comparison against Multi-Step Attacks

In Section 5.1, we compared the performance of single-step methods and observed that N-FGSM is able to match or surpass the state-of-the-art method, *i.e.,* GradAlign, while reducing the computational cost by a factor of 3. In this section, we compare the performance of N-FGSM against multi-step attacks. In particular, we compare against PGD-2 with $\alpha = {}^{\epsilon}\!/{}_2$ and PGD-10 with $\alpha = {}^2\!/{}_{255}$, keeping the same training settings as described in Section 5.1. PGD-x denotes x iterations and no restarts.

In Figure 4, we observe that PGD-2, despite being a multi-step method, still suffers from CO for larger $\epsilon$ as opposed to our proposed N-FGSM. On the other hand, despite achieving comparable clean accuracies, there is a gap in adversarial accuracies between PGD-10, and other single-step methods that grows with perturbation size. This can be partially expected since the search space for adversaries grows exponentially with $\epsilon$; and PGD, with more iterations, can explore it more thoroughly. Nevertheless, *computing a PGD-10 attack is $10\times$ more expensive to N-FGSM*. An important direction for future work would be addressing this gap and analysing, both theoretically and empirically, whether single-step methods can match the performance of their multi-step counterparts.

---

[3]https://github.com/tml-epfl/understanding-fast-adv-training/

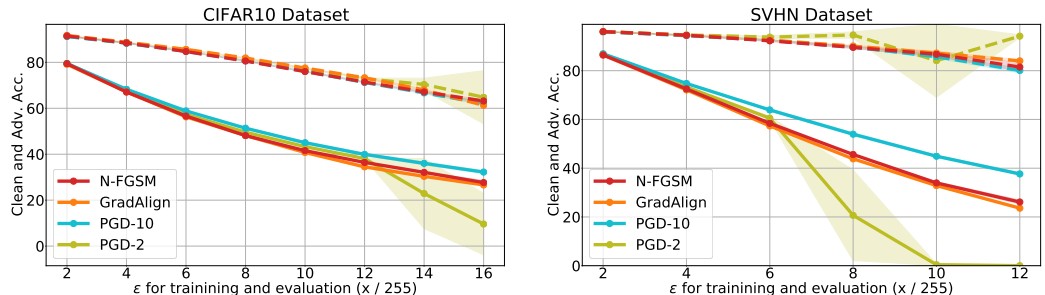

Figure 4: Comparison of N-FGSM and GradAlign with multi-step methods on CIFAR-10 (Left) and SVHN (Right) with PreactResNet18 over different perturbation radii ($\epsilon$ is divided by 255). Despite all methods achieving comparable clean accuracy (dashed lines), there is a gap in robust accuracy between PGD-10 and single-step methods. However, note that PGD-10 is $10\times$ more expensive than N-FGSM. Adversarial accuracy is based on PGD-50-10 and experiments are averaged over 3 seeds.

## 5.3 Analysis of Gradients and Adversarial Perturbations

To gain further insights into CO, we visually explore the perturbations generated with FGSM, RS-FGSM, N-FGSM, and PGD-10 attacks. We show that N-FGSM generates perturbations that exhibit behavior similar to PGD-10. In particular, for a given test sample, we average the adversarial perturbations ($\delta$) and gradients across several epochs at the beginning of training (Epoch 2 to 8) and at the end (Epoch 24 to 30) and visualise them in Figure 5 (see also Figure 12 in Appendix for more examples). We observe that, during early stages in training all, methods generate consistent and interpretable $\delta$. However, after CO, FGSM and RS-FGSM generate $\delta$ that are harder to interpret, similarly to their gradients. On the other hand, we observe that N-FGSM provides consistent and interpretable $\delta$ throughout training, similar to those generated by PGD-10. This provides further evidence that N-FGSM enjoys similar properties to the more expensive PGD-10 training.

Figure 5 analyzes the gradients and $\delta$ throughout the test set. Aside from loosing interpretability, post-CO the gradient norm increases by several orders of magnitude for FGSM and RS-FGSM while it remains low for N-FGSM and PGD-10. We also compute the effective rank[4] of $\delta$ for each example before and after CO to measure the consistency of $\delta$ before and after CO. We consider three training intervals, (Epoch 2 to 8): before CO for all methods; (Epoch 16 to 22): after FGSM suffers CO but not RS-FGSM; (Epoch 24 to 30): after both FGSM and RS-FGSM suffer CO. Prior to CO, PGD-10 has a larger effective rank (*i.e.,* the perturbations span a larger subspace) than FGSM and RS-FGSM. N-FGSM has the highest effective rank, arguably due to the higher noise magnitude. Note that RS-FGSM, which has a smaller noise magnitude and clipping, also has a larger effective rank than FGSM, however, the difference is much lower. When either FGSM or RS-FGSM suffer from CO, the effective rank of their $\delta$ increases significantly above that of PGD-10 and N-FGSM. This would suggest that $\delta$ loose consistency after CO and is aligned with our visualizations in Figure 5. All of these show properties of $\delta$ and gradients that are consistent across methods (N-FGSM and PGD) that avoid CO and different from methods like RS-FGSM and FGSM, which suffer from CO.

## 6 Increasing Adversarial Perturbations

In Section 4, we observed that removing clipping and increasing the noise magnitude were both necessary for the improved performance of N-FGSM. However, as discussed in Theorem N.2 this will result in an increase of the squared norm of the training perturbation $\delta_{\text{N}-\text{FGSM}}$ as compared to FGSM. In this section, we perform further ablations to corroborate that it is indeed the increase in noise magnitude – and not the mere increase of the perturbation's magnitude – that helps to stabilize N-FGSM.

**Increasing $\alpha$ alone is not sufficient.** N-FGSM combines a noise perturbation with an FGSM step. Thus, we can increase the perturbation magnitude by increasing any of the two. This begs the question: Is it sufficient to increase the N-FGSM step-size $\alpha$ to avoid CO without adding any noise? We observe in Figure 6 (A) that training without noise (essentially, FGSM) leads to CO, with robust accuracy

---

[4]We compute effective rank as the number of singular vectors required to explain $90\%$ of the variance.

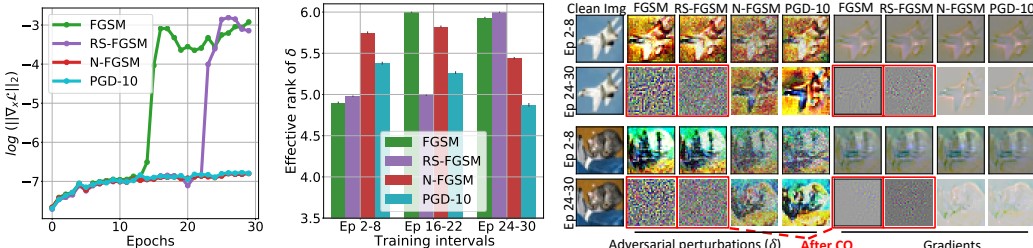

**Figure 5: Left:** Mean $\ell_2$ norm of per-sample-gradients across all test set samples. After CO, both FGSM and RS-FGSM gradients increase by several orders of magnitude. **Middle:** Effective rank of perturbations ($\delta$) across three training intervals *Ep 2-8* before CO for all methods; *Ep 16-22*: after FGSM presents CO but not RS-FGSM; *Ep 24-30*: after both FGSM and RS-FGSM had CO. **Right:** Visualization of $\delta$ and gradients averaged across several epochs at the beginning (top) and end (bottom) of training. After CO, FGSM and RS-FGSM obtain $\delta$ and gradients that are hard to interpret.

equal to zero, even for large values of $\alpha$. This indicates that it is not an increase in the perturbation norm, but the combination with noise which plays an essential role in circumventing CO for N-FGSM.

**Increasing $\alpha$ requires adjusting the noise magnitude.** As observed in Figure 6 (A), increasing $\alpha$ for N-FGSM leads to CO if the noise magnitude is not large enough. For example, while a noise magnitude $k = 1\epsilon$ and an adversarial step size $\alpha = 1.25\epsilon$ yield a robust accuracy of 49.68%, increasing $\alpha$ to $1.5\epsilon$ while keeping the same noise magnitude results in CO – with robust accuracy equal to zero. This further suggests that an increase in the adversarial step-size $\alpha$ requires a commensurate increase in the noise magnitude. We find that setting the noise magnitude $k = 2\epsilon$ works well for most settings.

**Larger noise perturbations preserve clean accuracy.** Increasing the norm of training perturbations by increasing $\alpha$ results in a drop in the clean accuracy (discussed later in Section 7). This has also been observed in prior works [35]. However, we show in Figure 4 that the clean accuracy for N-FGSM is similar to that of GradAlign, despite the magnitude of the perturbations being larger. We ablate the effects of adversarial and noise perturbations on the clean accuracy in Figure 6 (B): we observe that augmenting training samples with noise alone (*i.e.,* $\alpha = 0$) has a much milder effect on the clean accuracy than augmenting in an adversarial direction. In general, increasing noise is more forgiving on the clean accuracy than increasing the adversarial step size. This is not surprising, considering that moving in random directions along the input space has a significantly lower impact on the loss than moving along the FGSM direction (see Figure 14 in the Appendix) and that training with noise alone does not provide any significant robustness against larger attacks (for a more detailed ablation, see Appendix Figure 16).

**Other methods do not benefit from larger training $\epsilon$.** As previously mentioned, N-FGSM perturbations have $\ell_\infty$-norm larger than $\epsilon$. We have seen that the benefits of N-FGSM can not be reproduced by simply increasing $\alpha$ without increasing the noise. However, for the sake of completeness, we also ablate other single-step baselines by using a larger $\epsilon$ during training, while testing with a fixed $\epsilon = 8/255$ on CIFAR10. We observe that increasing $\epsilon_{\text{train}}$ seems to lead to a decrease in robustness for most methods; for instance, PGD-50-10 accuracy for RS-FGSM drops from $46.08 \pm 0.18$ when training with $\epsilon = 8/255$ to $0.0 \pm 0.0$ with $\epsilon = 12/255$. In two cases (GradAlign and MultiGrad), we observe a small increase, with the highest increase being for GradAlign, which improves from $48.14 \pm 0.15$ to $50.6 \pm 0.45$; yet, the clean accuracy drops from $81.9 \pm 0.22$ to $73.29 \pm 0.23$. This is similar to increasing $\alpha$ for N-FGSM (see Figure 6 (C)). However, this is tied to a significant degradation of clean accuracy. All in all, taking into account both clean and robust accuracy, we conclude that all single-step baselines suffer from either CO or a severe degradation in their clean accuracy when increasing the training $\epsilon$. Full results are presented in Table 2 in Appendix.

## 7 Additional Ablations

**Hyperparameter selection.** While FGSM relies on a fixed step-size (*i.e.,* $\alpha = \epsilon$), Wong et al. [35] explored different values of $\alpha$ for RS-FGSM, finding that an increase of the step-size improves the adversarial accuracy – up to a point where CO occurs. We also ablate the value of $\alpha$ for N-FGSM in Figure 6 (C). We find that by increasing the noise magnitude, N-FGSM can use larger $\alpha$ values than RS-

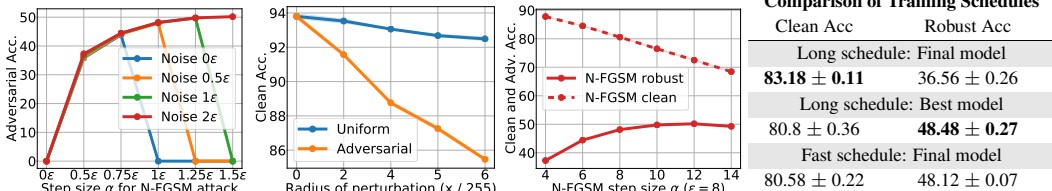

Figure 6: Different ablations on N-FGSM parameters and training schedule. From left to right: **A:** Adversarial accuracy when varying step-size $\alpha$ and noise magnitude $k$ ($\epsilon = 8$). Increasing $\alpha$ does not suffice to prevent CO, we must also increase the noise magnitude. **B:** Clean accuracy after training with random or adversarial perturbations. With comparable radius, random perturbations have a much milder effect than adversarial. **C:** Ablation of step size $\alpha$ in N-FGSM $\epsilon = 8, k = 2\epsilon$. As we increase the magnitude of the FGSM perturbation we observe an increase in robustness coupled with a drop on the clean accuracy. **D**: Comparison of the "fast" training schedule from [35] and "long" training schedule described in [24]. N-FGSM shows robust oberfitting but not CO with the long schedule. Adversarial accuracy is based on PGD-50-10 and experiments are averaged over 3 seeds.

FGSM, without suffering from CO. This leads to an increase in the adversarial accuracy at the expense of a decrease in the clean accuracy. In light of this trade-off, we also use $\alpha = \epsilon$ for N-FGSM. Regarding the noise hyperparameter $k$, we find that $k = 2\epsilon$ works in all but one SVHN experiment ($\epsilon = 12$, in which we set $k = 3\epsilon$). In comparison, GradAlign regularizer hyperparameter or ZeroGrad quantile value need to be modified for every radius with a noticeable shift between CIFAR-10 and SVHN hyperparameters, suggesting they may require additional tuning when applied to novel datasets.

**Long vs fast training schedules.** Throughout our experiments, we used the RS-FGSM training setting introduced in [35]. However, Rice et al. [24] suggest that a longer training schedule coupled with early stopping may lead to a boost in performance. Kim et al. [15] and Li et al. [19] report that longer training schedules increase the chances of CO for RS-FGSM and that this limits its performance. We test the longer training schedule with N-FGSM and find that it does not suffer from CO. However, it does suffer from *robust overfitting*, *i.e.,* adversarial accuracy on the training set is larger than on the test set as described in [24] for PGD-10. Notice the difference between the robust accuracy of the final and best models in Figure 6 (D). Interestingly, although we observe a slight increase in performance when using the long training schedule, we find the fast training schedule to be remarkably competitive. See more results in Appendix D, including a comparison to GradAlign.

**Experiments with WideResNet28-10.** We also compare the performance of all methods on WideResNet28-10 [36] architecture in Figure 8 and Figure 9 in Appendix. As in the experiments with PreActResNet18, N-FGSM obtains the best performance/cost trade-off. We had to increase the regularizer hyperparameter for GradAlign (compared to the settings for PreActResNet18) in order to prevent CO on CIFAR-100 and, to our surprise, *we could not find a competitive hyperparameter setting for GradAlign on the SVHN dataset* for $\epsilon \geq 6$. We tried both increasing the regularizer hyperparameter and decreasing the step size $\alpha$, but some or all runs led to models close to a constant classifier for each setting. We do not claim that GradAlign will not work, but finding a good configuration might require further tuning. The default configuration for N-FGSM ($\alpha = \epsilon$, $k = 2\epsilon$) works well in all settings except for $\epsilon = 16$ on CIFAR-10 and $\epsilon = 10, 12$ on SVHN. For CIFAR-10, we increase the noise magnitude to $k = 4\epsilon$. For SVHN, we find that decreasing $\alpha$ works better than increasing the noise. In both cases, N-FGSM yields nontrivial adversarial accuracy.

**Experiments on Imagenet.** We present results on the Imagenet dataset [17] in Table 1. Due to the high computational demands of Imagenet training and testing we focus on the main baselines of comparable cost to FGSM. Namely FGSM, RS-FGSM and N-FGSM. We observe that FGSM presents CO for $\epsilon = {}^6/_{255}$ while neither RS-FGSM nor N-FGSM present CO. However, N-FGSM has better robustness. For instance, at $\epsilon = {}^6/_{255}$ N-FGSM obtains PGD50-10 accuracy of 17.12% while RS-FGSM yields 16.5% and FGSM 0.08% (due to CO). Thus, N-FGSM also avoids CO in ImageNet, improving robustness over same-cost baselines. For experimental details refer to Appendix L.

**Combining N-FGSM with additional regularizers.** Recent works [29, 30] have been proposed to improve performance of single step methods at moderate perturbation radii $\epsilon = {}^8/_{255}$. However, we observe that with the default settings (which use a version of RS-FGSM with Bernoulli noise) they lead to CO for larger $\epsilon$. Then we compare them with N-FGSM + Regularizer where we apply

Table 1: Clean accuracy (top) and PGD50-10 accuracy (bottom) of N-FGSM and other same-cost baselines on Imagenet dataset. We observe that FGSM presents CO for $\epsilon = {}^6/_{255}$ while both RS-FGSM and N-FGSM avoid CO. N-FGSM has consistently better robustness than baselines.

|  | $\epsilon = {}^2/_{255}$ | $\epsilon = {}^4/_{255}$ | $\epsilon = {}^6/_{255}$ |
|---|---|---|---|
| FGSM | 54.72 | 48.50 | 48.55 |
|  | 38.21 | 25.86 | 0.08 |
| RS-FGSM | 56.29 | 50.81 | 47.67 |
|  | 36.86 | 25.12 | 16.49 |
| N-FGSM | 54.39 | 47.56 | 47.70 |
|  | 38.07 | 26.28 | 17.12 |

their proposed regularizers to N-FGSM. If we apply GAT or NuAT regularizers to N-FGSM then we do not observe CO and usually a boost in performance. For instance, at $\epsilon = {}^{10}/_{255}$, GAT has a robust acc (with PGD50-10) of $43.34 \pm 0.23$ while N-FGSM+GAT regularizer obtains $44.97 \pm 0.07$, in comparison plain N-FGSM has $41.56 \pm 0.16$. This is extremely compelling as it suggests N-FGSM can be combined with other regularizers designed to improve FGSM performance and mutually benefit each other. Full results of the comparison are presented in Table 6 in Appendix K

## 8 Conclusion

In this work, we explore the role of noise and clipping in single-step adversarial training. Contrary to previous intuitions, we show that increasing the noise magnitude and removing the $\epsilon - \ell_\infty$ constraint leads to an improvement in adversarial robustness while maintaining a competitive clean accuracy. These findings led us to propose N-FGSM, a simple and effective approach that can match or surpass the performance of GradAlign [1], while achieving a $3\times$ speed-up.

We perform an extensive comparison with other relevant single-step methods, observing that all of them achieve sub-optimal performance and most of them are not able to avoid CO for larger $\epsilon$ attacks. Moreover, we also analyze gradients and adversarial perturbations during training and observe that they have a similar behaviour for N-FGSM and PGD-10 as opposed to other methods that present CO such as FGSM and RS-FGSM. However, despite impressive improvements of single-step adversarial training methods, there is still a gap between single-step and multi-step methods such as PGD-10 as we increase the $\epsilon$ radius. Therefore, future work should put an emphasis on formally understanding the limitations of single-step adversarial training and explore how, if possible, this gap can be reduced.

### Acknowledgments and Disclosure of Funding

We thank Guillermo Ortiz-Jiménez for the fruitful discussions and feedback. This work is supported by the UKRI grant: Turing AI Fellowship EP/W002981/1 and EPSRC/MURI grant: EP/N019474/1. We would also like to thank the Royal Academy of Engineering and FiveAI. A. Sanyal acknowledges support from the ETH AI Center postdoctoral fellowship. Pau de Jorge is fully funded by NAVER LABS Europe.

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
