# A Additional plots for PreActResNet18 experiments

In the main paper we compare N-FGSM with other single-step methods and multi-step methods separately and remove clean accuracies for better visualization. In this section we present the curves for all methods with both the clean and robust accuracy. The tendency in the three datasets is for N-FGSM PGD-50-10 accuracy to be slightly above that of GradAlign, while the opposite happens to the clean accuracy. We also observe that clean accuracy becomes significantly more noisy when CO happens. Exact numbers for all the curves are in Appendix S.

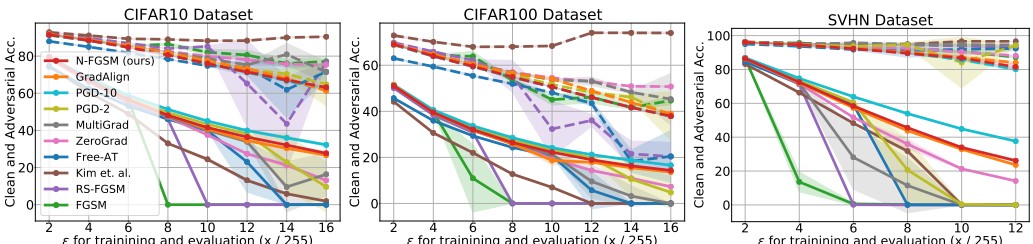

Figure 7: Comparison of all methods on CIFAR-10, CIFAR-100 and SVHN with PreactResNet18 over different perturbation radius ($\epsilon$ is divided by 255). We plot both the robust (solid line) and the clean (dashed line) accuracy for each method. Our method, N-FGSM, is able to match or surpass the state-of-the-art single-step method GradAlign while *reducing the cost by a $3\times$ factor*. Adversarial accuracy is based on PGD-50-10 and experiments are averaged over 3 seeds. Legend is shared among all plots.

# B Experiments with WideResNet28-10 architecture

In this section we present the plots of our experiments with WideResNet28-10. We report the results in two figures. In Figure 8 we compare all single-step methods and we do not plot the clean accuracy for better visualization. In Figure 9 we plot all methods, including multi-step methods, and report the clean accuracy as well with dashed lines. Since we observed that our baseline, RandAlpha, outperformed [15] in all settings for PreActResNet18, we only report RandAlpha for WideResNet. As mentioned in the main paper, we observe that CO seems to be more difficult to prevent for WideResNet. In particular, for GradAlign we observed the regularizer hyperparameter settings proposed by [1] for CIFAR-10 (searched for a PreActResNet18) worked well. However, those parameters led to CO for $6 \le \epsilon \le 12$ in CIFAR-100. Since $\epsilon = 14, 16$ did not show CO, we increased the GradAlign regularizer hyperparameter $\lambda$ for CIFAR-100 so that each $6 \le \epsilon \le 12$ would have the default value corresponding to $\epsilon + 2$, for instance, $\lambda$ for $\epsilon = 6$ would be the default $\lambda$ in [1] for $\epsilon = 8$.

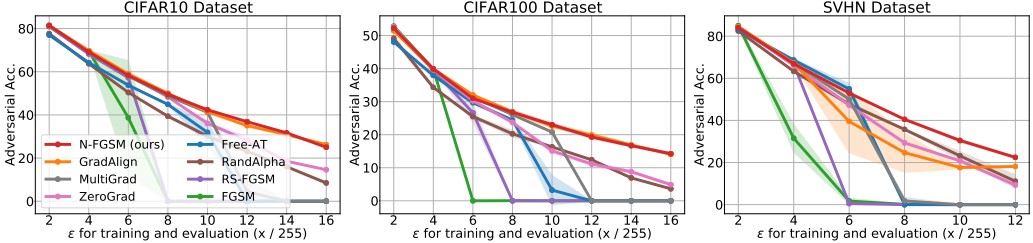

Figure 8: Comparison of single-step methods on CIFAR-10, CIFAR-100 and SVHN with WideResNet28-10 over different perturbation radius ($\epsilon$ is divided by 255). Our method, N-FGSM, is able to match or surpass the state-of-the-art single-step method GradAlign while *reducing the cost by a $3\times$ factor*. Moreover, we could not find any competitive hyperparameter setting for GradAlign for $\epsilon \ge 6$ in SVHN dataset. Adversarial accuracy is based on PGD-50-10 and experiments are averaged over 3 seeds. Legend is shared among all plots.

For SVHN we observed that the default values for $\lambda$ led to models close to a constant classifier for $\epsilon \ge 6$. We tried to increase the lambda for those $\epsilon$ values to $1.25\lambda$ but observed the same result. Since

the model did not show typical CO but rather it seemed as it was underfitting, we tried to reduce the step-size to $\alpha = 0.75\epsilon$ and also both decreasing $\alpha$ and increasing $\lambda$. When reducing the step size we obtain accuracies above those of a constant classifier for some radii, however, some or all seeds converge to a constant classifier for each setting, hence the large standard deviations. For N-FGSM, the default configuration of N-FGSM ($\alpha = \epsilon$, $k = 2\epsilon$) works well in all settings except for $\epsilon = 16$ on CIFAR-10 and $\epsilon = 10$, $12$ on SVHN. For CIFAR-10, we increase the noise magnitude to $k = 4\epsilon$. For SVHN we find that decreasing $\alpha$ as we tried for GradAlign works better than increasing the noise. We use $\alpha = 8$ for both $\epsilon$ radii. Exact numbers for all the curves are in Appendix S

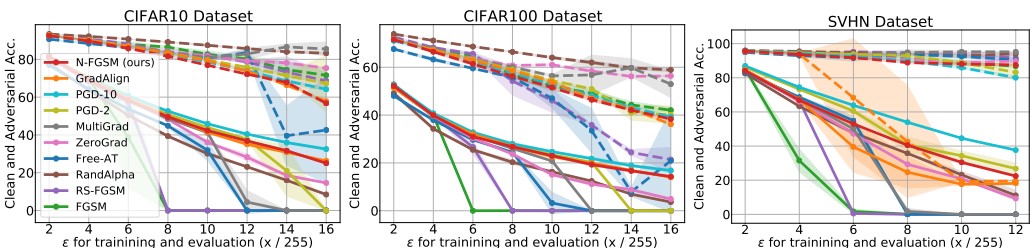

Figure 9: Comparison of all methods on CIFAR-10, CIFAR-100 and SVHN with WideResNet28-10 over different perturbation radius ($\epsilon$ is divided by 255). We plot both the robust (solid line) and the clean (dashed line) accuracy for each method. Legend is shared among all plots.

## C   Increasing adversarial perturbations during training

As mentioned in the main paper, N-FGSM perturbations have $\ell_\infty-$norm larger than $\epsilon$, see Appendix N. In Section 6 we have seen that the benefits of N-FGSM can not be reproduced by simply increasing $\alpha$ without increasing the noise. However, for the sake of completeness, we also ablate other single-step baselines by using a larger $\epsilon$ during training *i.e.,* $\{\epsilon = 8/255, \epsilon = 12/255, \epsilon = 16/255\}$ while testing with a fixed $\epsilon = 8/255$ on CIFAR-10. Results are presented in Table 2. We observe that increasing $\epsilon_{\text{train}}$ seems to lead to a decrease in robustness for most methods, *e.g.,* PGD-50-10 accuracy for RS-FGSM goes from $46.08 \pm 0.18$ when training with $\epsilon = 8/255$ to $0.0 \pm 0.0$ with $\epsilon = 12/255$. In two cases (GradAlign and MultiGrad) we observe a small increase, highest increase is for GradAlign which goes from $48.14 \pm 0.15$ to $50.6 \pm 0.45$, however, the clean accuracy drops from $81.9 \pm 0.22$ to $73.29 \pm 0.23$. This is similar to increasing $\alpha$ for N-FGSM (see Figure 6 (C)). However, this is tied to a significant degradation of clean accuracy. All in all, taking into account both clean and robust accuracy we conclude all baselines perform best without increasing the training $\epsilon$. All ablation results are presented in Table 2.

Table 2: Ablation of the PGD-50-10 accuracy for single-step methods when increasing the $\epsilon_{\text{train}}$. All models are evaluated with PGD-50-10 attack and $\epsilon_{\text{test}} = 8/255$. Note that considering the trade-off between clean and robust accuracy, all methods perform best when training with the same epsilon to be applied at test time.

| Method | $\epsilon_{\text{train}} = 1\epsilon_{\text{test}}$ | | $\epsilon_{\text{train}} = 1.5\epsilon_{\text{test}}$ | | $\epsilon_{\text{train}} = 2\epsilon_{\text{test}}$ | | |
|---|---|---|---|---|---|---|---|
| | **Clean acc.** | **PGD acc.** | **Clean acc.** | **PGD acc.** | **Clean acc.** | **PGD acc.** | **Rel. Cost** |
| GradAlign | $81.9 \pm 0.22$ | $48.14 \pm 0.15$ | $73.29 \pm 0.23$ | $50.6 \pm 0.45$ | $61.3 \pm 0.15$ | $46.67 \pm 0.29$ | 3 |
| MultiGrad | $82.33 \pm 0.14$ | $47.29 \pm 0.07$ | $75.28 \pm 0.2$ | $50.0 \pm 0.79$ | $71.42 \pm 5.63$ | $0.0 \pm 0.0$ | 2 |
| AT Free | $78.41 \pm 0.18$ | $46.03 \pm 0.36$ | $73.91 \pm 4.19$ | $32.4 \pm 22.91$ | $71.64 \pm 3.89$ | $0.0 \pm 0.0$ | 1.6 |
| Kim et. al. | $89.02 \pm 0.1$ | $33.01 \pm 0.09$ | $88.35 \pm 0.31$ | $27.36 \pm 0.31$ | $90.45 \pm 0.08$ | $9.28 \pm 0.12$ | 1.5 |
| FGSM | $86.41 \pm 0.7$ | $0.0 \pm 0.0$ | $80.6 \pm 2.59$ | $0.0 \pm 0.0$ | $77.14 \pm 2.46$ | $0.0 \pm 0.0$ | 1 |
| RS-FGSM | $84.05 \pm 0.13$ | $46.08 \pm 0.18$ | $65.22 \pm 23.23$ | $0.0 \pm 0.0$ | $76.66 \pm 0.38$ | $0.0 \pm 0.0$ | 1 |
| ZeroGrad | $82.62 \pm 0.05$ | $47.08 \pm 0.1$ | $78.11 \pm 0.2$ | $46.43 \pm 0.37$ | $75.42 \pm 0.13$ | $45.63 \pm 0.39$ | 1 |
| N-FGSM | $80.58 \pm 0.22$ | $48.12 \pm 0.07$ | $71.46 \pm 0.14$ | $50.23 \pm 0.31$ | $63.18 \pm 0.49$ | $46.46 \pm 0.1$ | 1 |

# D   Longer training schedule

In our experiments, we have followed the "fast" training schedule introduced by [35]. However, [24] suggest that a longer training schedule coupled with early stopping may lead to a boost in performance. We also use the long training schedule for N-FGSM and observe that it does not lead to CO. In Table 3 we compare the performance of N-FGSM and GradAlign for the long training schedule. We observe that GradAlign does not seem to benefit from the long training schedule. On the other hand, although N-FGSM seems to obtain a slight increase in performance, the "fast" schedule provides comparable performance. It is worth mentioning that for GradAlign, the default regularizer hyperparameter for $\epsilon = {}^8/_{255}$ and CIFAR-10 ($\lambda = 0.2$) does not prevent CO. We do a hyperparameter search and keep the value with the largest final robust accuracy ($\lambda = 0.632$).

Table 3: Comparison of "long" [24] and "fast" [35] training schedules for N-FGSM and GradAlign. GradAlign does not seem to benefit from the long training schedule. Although N-FGSM seems to obtain a slight increase in performance, the "fast" schedule provides comparable performance.

| N-FGSM | | Grad Align | |
|---|---|---|---|
| Clean Acc | Robust Acc | Clean Acc | Robust Acc |
| **Long schedule: Final model** | | | |
| $\mathbf{83.18 \pm 0.11}$ | $36.56 \pm 0.26$ | $\mathbf{84.13 \pm 0.24}$ | $36.17 \pm 0.19$ |
| **Long schedule: Best model** | | | |
| $80.8 \pm 0.36$ | $\mathbf{48.48 \pm 0.27}$ | $81.57 \pm 0.44$ | $47.86 \pm 0.1$ |
| **fast schedule: Final model** | | | |
| $80.58 \pm 0.22$ | $48.12 \pm 0.07$ | $81.9 \pm 0.22$ | $\mathbf{48.14 \pm 0.15}$ |

In Table 3 we observe that the performance of the final model is lower than that of an early stopped method. This could be expected due to the phenomena of robust overfitting described in [24]. However, as a sanity check to make sure that this is not due to a hidden CO during the long schedule which somehow the model recovers from we plot the full training history in Figure 10. There we can observe that *for N-FGSM there is no CO during training*.We also show FGSM (which is well known has CO for $\epsilon = {}^8/_{255}$) for comparison.

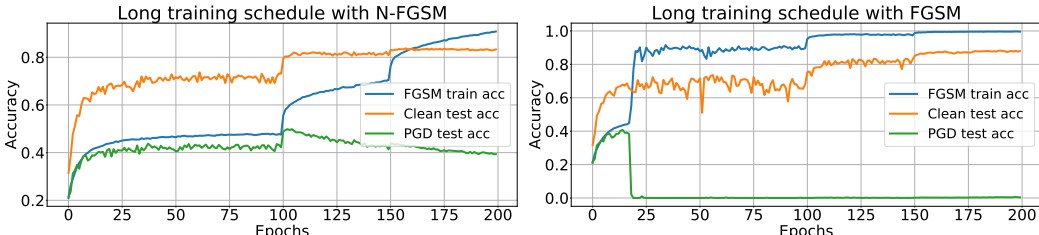

Figure 10: Training and test accuracy during the long training schedule proposed in [24]. We observe that N-FGSM (left) does not present CO at any point during training, however suffers from robust overfitting as described in [24] which suggested selecting the best validated model as a simple and yet effective way to improve robustness. On the other hand FGSM (right) suffers from CO where the robustness drops suddenly to 0 and does not recover.

# E   Randomized Alpha

Kim et al. [15] evaluate intermediate points along the RS-FGSM direction in order to pick the "optimal" perturbation size. However, we find that increasing the number of intermediate evaluated points does not necessarily lead to increased adversarial accuracy. Moreover, for large perturbations we could not prevent CO even with twice the number of evaluations tested by [15]. This motivates

us to test a very simple baseline where instead of evaluating intermediate steps, the RS-FGSM perturbation size is randomly selected as: $\delta = t \cdot \delta_{\text{RS-FGSM}}$ where $t \sim \mathcal{U}[0, 1]^d$. Interestingly, as reported in Figure 11, we find that this very simple baseline, dubbed *RandAlpha*, is able to avoid CO for all values of $\epsilon$ and outperforms [15] on CIFAR-10, CIFAR-100 and SVHN. This is aligned with our main finding that combining noise with adversarial attacks is indeed a powerful tool that should be explored more thoroughly before developing more expensive solutions.

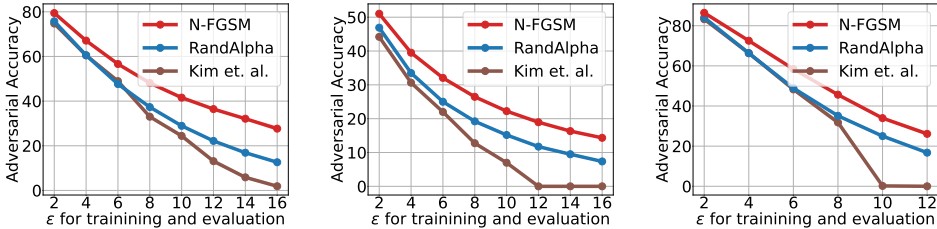

Figure 11: Comparison of [15] with RandomAlpha, our baseline where we multiply the RS-FGSM perturbation by a scalar uniformly sampled in $[0, 1]$. We present results on CIFAR-10 (Left), CIFAR-100 (Middle) and SVHN (Right) with PreActResNet18.

# F Further visualizations of adversarial perturbations and gradients

In this section we present an extension of Figure 5 with further examples. As observed in the main paper, early in training adversarial perturbations ($\delta$) and gradients are consistent across epochs, however, after CO they become hard to interpret. Note that although we label rows as either pre-CO or post-CO we only observe CO for FGSM and RS-FGSM. Both PGD-10 and N-FGSM obtain robust models as shown in detail in the paper.

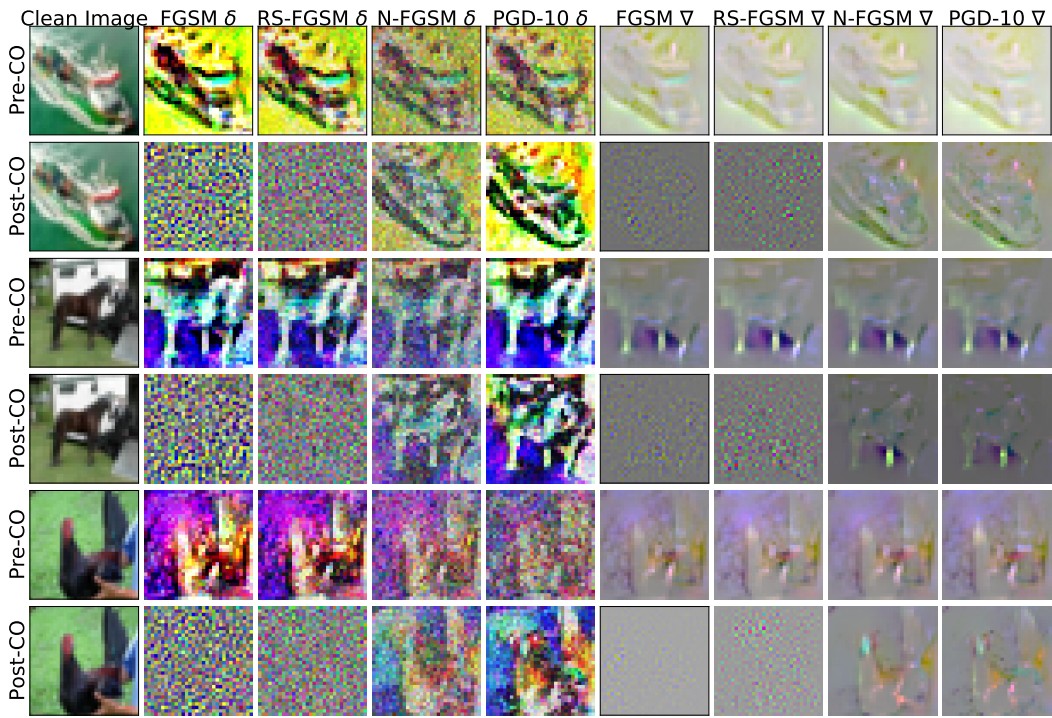

Figure 12: Visualization of adversarial perturbations ($\delta$'s) and gradients averaged across several epochs before CO (pre-CO) and after (post-CO). Note that only FGSM and RS-FGSM present CO, PGD-10 and N-FGSM do not. Post-CO, FGSM and RS-FGSM obtain $\delta$'s that are hard to interpret, idem for their gradients.

# G Robust evaluations with autoattack

Table 4: Clean (top) and robust accuracy (bottom) for CIFAR-10 and PreacResNet18 evaluated with autoattack (AA) [6]. We find the same trend as with PGD50-10.

|  | $\epsilon = 2/255$ | $\epsilon = 4/255$ | $\epsilon = 6/255$ | $\epsilon = 8/255$ | $\epsilon = 10/255$ | $\epsilon = 12/255$ | $\epsilon = 14/255$ | $\epsilon = 16/255$ |
|---|---|---|---|---|---|---|---|---|
| FGSM | $91.52 \pm 0.08$ | $88.59 \pm 0.08$ | $85.17 \pm 0.03$ | $86.62 \pm 0.08$ | $83.35 \pm 2.03$ | $78.51 \pm 3.3$ | $77.31 \pm 1.9$ | $75.88 \pm 1.49$ |
|  | $78.99 \pm 0.19$ | $65.99 \pm 0.24$ | $54.0 \pm 0.32$ | $0.0 \pm 0.0$ | $0.0 \pm 0.0$ | $0.0 \pm 0.0$ | $0.0 \pm 0.0$ | $0.0 \pm 0.0$ |
| GradAlign | $91.48 \pm 0.08$ | $88.55 \pm 0.18$ | $85.23 \pm 0.22$ | $81.69 \pm 0.1$ | $77.73 \pm 0.18$ | $73.46 \pm 0.16$ | $67.87 \pm 0.5$ | $61.66 \pm 0.32$ |
|  | $79.09 \pm 0.21$ | $65.65 \pm 0.13$ | $53.99 \pm 0.2$ | $44.11 \pm 0.34$ | $35.72 \pm 0.34$ | $28.66 \pm 0.15$ | $22.93 \pm 0.33$ | $18.4 \pm 0.28$ |
| N-FGSM | $91.44 \pm 0.09$ | $88.36 \pm 0.04$ | $84.56 \pm 0.12$ | $80.36 \pm 0.03$ | $75.81 \pm 0.22$ | $71.03 \pm 0.16$ | $66.49 \pm 0.36$ | $62.86 \pm 0.88$ |
|  | $78.99 \pm 0.17$ | $66.06 \pm 0.25$ | $53.94 \pm 0.3$ | $44.36 \pm 0.26$ | $36.73 \pm 0.27$ | $30.45 \pm 0.2$ | $25.08 \pm 0.15$ | $19.0 \pm 1.08$ |

Following previous work, [1, 11] we have evaluated robustness with PGD50-10, i.e. PGD with 50 iterations and 10 restarts. However, for the sake of completeness, we also present results of robust accuracy evaluated with autoattack [6]. In Table 4 we evaluate models adversarially trained with our proposed method N-FGSM, the baseline FGSM and GradAlign. We observe the same pattern as with the PGD50-10 attack, therefore we are convinced that our results are general.

## H Catastrophic Overfitting outside the ResNet family

Previous work focusing on CO has only used architectures from the ResNet family. In Table 5 we present results for adversarial training with a VGG-16 architecture [28]. Similarly to other studied models we observe that FGSM leads to CO while N-FGSM is able to prevent it. However, it seems that FGSM presents CO for slighly larger $\epsilon$ radii, indicating that the architecture might play a role in CO. We consider investigating this further a promising direction of future work.

Table 5: Clean (top) and robust accuracy (bottom) for CIFAR-10 and VGG-16 [28] evaluated with PGD50-10. We also observe CO for VGG architecture when trained with FGSM, moreover, N-FGSM is able to prevent CO. Interestingly, for VGG CO happens for slighly large $\epsilon$ values indicating that the architecture might play a role in CO.

|  | $\epsilon = {}^4/_{255}$ | $\epsilon = {}^6/_{255}$ | $\epsilon = {}^8/_{255}$ | $\epsilon = {}^{10}/_{255}$ | $\epsilon = {}^{12}/_{255}$ | $\epsilon = {}^{14}/_{255}$ | $\epsilon = {}^{16}/_{255}$ |
|---|---|---|---|---|---|---|---|
| FGSM | $85.04 \pm 0.1$ | $79.34 \pm 0.11$ | $73.39 \pm 0.0$ | $82.6 \pm 0.0$ | $83.04 \pm 0.0$ | $81.4 \pm 0.0$ | $80.41 \pm 0.21$ |
|  | $62.94 \pm 0.07$ | $52.72 \pm 0.12$ | $44.0 \pm 0.02$ | $0.07 \pm 0.0$ | $0.8 \pm 0.0$ | $0.25 \pm 0.0$ | $0.31 \pm 0.15$ |
| N-FGSM | $84.53 \pm 0.0$ | $79.42 \pm 0.0$ | $72.01 \pm 0.28$ | $66.81 \pm 0.54$ | $61.19 \pm 0.0$ | $56.97 \pm 0.0$ | $53.1 \pm 1.19$ |
|  | $63.32 \pm 0.0$ | $53.0 \pm 0.0$ | $44.3 \pm 0.09$ | $38.25 \pm 0.1$ | $33.36 \pm 0.0$ | $29.23 \pm 0.0$ | $25.72 \pm 0.22$ |

## I Further increasing the attack radii

Following previous work [1] we have studied $\epsilon$ attack radii up to $epsilon = {}^{16}/_{255}$. Indeed, the performance at these radius is already significantly degraded and thus it would not be very practical for most applications. However, to show that N-FGSM can prevent CO at even larger radii we test two additional radii, $\epsilon = {}^{20}/_{255}$ and $\epsilon = {}^{24}/_{255}$. In both cases N-FGSM is able to prevent CO. For $\epsilon = {}^{20}/_{255}$ we obtain a clean accuracy of $51.63 \pm 0.38$ and robust of $20.62 \pm 0.37$ while for $\epsilon = {}^{24}/_{255}$ we obtain a clean accuracy of $40.16 \pm 0.96$ and robust of $15.3 \pm 1.49$. We argue that it is of little interest to try even larger perturbations unless more effective methods to improve both the clean and robust performance are found.

## J Testing other norms

Following previous work, we have focused on the $\ell_\infty$ threat model. Although this is where works studying CO have mainly focused, we observe that CO is also present in other norms such as $\ell_1$ and $\ell_2$. Moreover, in both cases we observe that N-FGSM is able to prevent CO. Interestingly, the range of norms in which we observe CO is usually much higher than normally tested for these norms which would explain why the $\ell_\infty$ norm has been the main focus of study in related works.

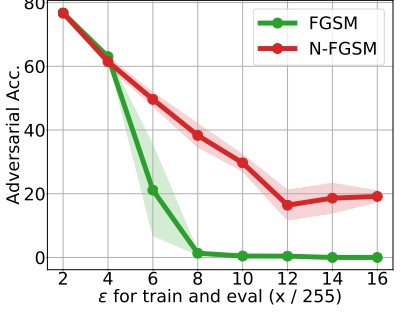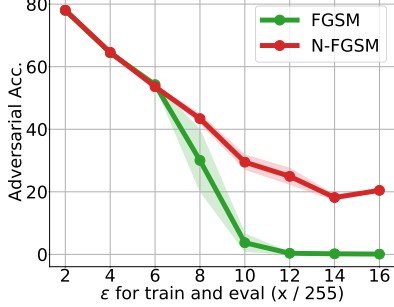

Figure 13: Robust accuracy after training with FGSM or N-FGSM using $\ell_1$ (left) and $\ell_2$ (right) perturbations. As observed for $\ell_\infty$ perturbations FGSM leads to CO, while N-FGSM is able to prevent it. Note that the strength of the perturbations is indicated to be equivalent to $\ell_\infty$ perturbations where all pixels have maximum magnitude i.e. $\epsilon = 8/255$ indicates perturbations were restricted to an $\ell_p$ norm of a vector where all components are in $\{-\epsilon, +\epsilon\}$. Which would correspond to an $\ell_1$ norm of $n\epsilon$ and an $\ell_2$ norm of $\epsilon\sqrt{n}$ where $n$ indicates the dimensionality of the input.

## K    Combining N-FGSM with additional regularizers

In this section we present the results from Section 7 where we combine N-FGSM with additional regularizers [29, 30] that were proposed for single-step adversarial training to boost the performance. First, we try the proposed methods with the default settings (which use a version of RS-FGSM with Bernoulli noise) and observe they lead to CO for larger $\epsilon$. Then we compare them with N-FGSM + Regularizer where we apply their proposed regularizers to N-FGSM. If we apply GAT or NuAT regularizers to N-FGSM then we do not observe CO and usually a boost in performance. Results are presented in Table 6.

Table 6: Clean accuracy (top) and PGD50-10 accuracy (bottom) of N-FGSM with additional regularizers introduced in GAT [29] and NuAT [30]. Both GAT and NuAT present CO with their default training method. If we apply their proposed regularizers to N-FGSM we can avoid CO while achieving a boost in performance.

|  | $\epsilon = 2/255$ | $\epsilon = 4/255$ | $\epsilon = 6/255$ | $\epsilon = 8/255$ | $\epsilon = 10/255$ | $\epsilon = 12/255$ | $\epsilon = 14/255$ | $\epsilon = 16/255$ |
|---|---|---|---|---|---|---|---|---|
| GAT | $88.79 \pm 0.15$ | $84.35 \pm 0.11$ | $80.16 \pm 0.15$ | $76.75 \pm 0.38$ | $73.71 \pm 0.12$ | $80.44 \pm 5.08$ | $83.9 \pm 1.0$ | $82.17 \pm 2.47$ |
|  | $80.04 \pm 0.06$ | $68.51 \pm 0.08$ | $59.16 \pm 0.24$ | $50.98 \pm 0.12$ | $43.34 \pm 0.23$ | $14.93 \pm 9.26$ | $2.33 \pm 0.58$ | $1.25 \pm 0.51$ |
| N-FGSM+GAT | $89.1 \pm 0.08$ | $84.84 \pm 0.05$ | $81.38 \pm 0.07$ | $78.28 \pm 0.04$ | $75.66 \pm 0.35$ | $73.56 \pm 0.23$ | $70.84 \pm 0.51$ | $65.48 \pm 0.96$ |
|  | $79.96 \pm 0.21$ | $69.5 \pm 0.18$ | $60.06 \pm 0.09$ | $51.8 \pm 0.34$ | $44.97 \pm 0.07$ | $38.71 \pm 0.16$ | $32.71 \pm 0.11$ | $27.87 \pm 0.35$ |
| NuAT | $87.81 \pm 0.24$ | $82.9 \pm 0.18$ | $78.06 \pm 0.2$ | $73.22 \pm 0.34$ | $71.08 \pm 4.87$ | $74.38 \pm 7.32$ | $78.5 \pm 1.54$ | $80.1 \pm 1.08$ |
|  | $79.49 \pm 0.03$ | $67.77 \pm 0.13$ | $57.93 \pm 0.17$ | $50.1 \pm 0.33$ | $34.35 \pm 9.0$ | $17.54 \pm 8.82$ | $6.6 \pm 0.77$ | $3.29 \pm 0.87$ |
| NGFSM+NuAT | $87.92 \pm 0.0$ | $83.54 \pm 0.0$ | $78.86 \pm 0.25$ | $74.61 \pm 0.34$ | $70.37 \pm 0.12$ | $65.56 \pm 0.19$ | $60.76 \pm 0.74$ | $52.79 \pm 0.66$ |
|  | $79.52 \pm 0.0$ | $68.36 \pm 0.0$ | $58.88 \pm 0.16$ | $51.12 \pm 0.2$ | $44.62 \pm 0.38$ | $38.24 \pm 0.38$ | $32.85 \pm 0.58$ | $29.19 \pm 0.35$ |
| N-FGSM | $91.48 \pm 0.17$ | $88.44 \pm 0.09$ | $84.72 \pm 0.04$ | $80.58 \pm 0.22$ | $75.98 \pm 0.1$ | $71.46 \pm 0.14$ | $67.11 \pm 0.37$ | $63.18 \pm 0.49$ |
|  | $79.43 \pm 0.21$ | $67.09 \pm 0.31$ | $56.62 \pm 0.26$ | $48.12 \pm 0.07$ | $41.56 \pm 0.16$ | $36.43 \pm 0.16$ | $32.11 \pm 0.2$ | $27.67 \pm 0.93$ |

## L    Imagenet experimental details

For our experiments on Imagenet we mainly follow the settings from [35]. However, for simplicity we did not do image resizing which requires storing two additional Imagenet datasets. More importantly, we found that the learning rate schedule suggested in [35] was not optimal for N-FGSM. The schedule suggested in [35] follows three different stages in which the learning increases or decreases linearly for some iterations. In particular in the first stage, the learning rate has an initial warm-up where it increases linearly from 0.0 to 0.4 during the first epoch and then decreases linearly to 0.04 during the next 5 epochs. As a lucky coincidence when debugging, we modified this initial stage such that we preserved the initial increase to 0.4 for the first epoch, but then we directly jumped to a learning rate of 0.04 which remained constant for the next 5 epochs. For phase 2 and 3 both schedules remained the same. First decreasing from 0.04 to 0.004 for epoch 6 to 12 and finally from 0.004 to 0.0004 for epoch 12 to 15. This small change made N-FGSM improve both in clean and robust accuracy for $\epsilon = 4/255$, $6/255$ and these are the numbers reported. This indicates that further tuning the learning rate schedule might be an effective way to improve performance and even help prevent CO, however, due to the computational demands of ImageNet adversarial training we leave it for future work. To be thorough we also trained RS-FGSM and FGSM with the modified schedule and found that neither of them benefit from it. Regarding N-FGSM hyperparameters, for $\epsilon = 2/255$ we used $\alpha = 2/255$ and $k = 1$; for $\epsilon = 4/255$ we used $\alpha = 4/255$ and $k = 1$; and for $\epsilon = 6/255$ we also used $\alpha = 4/255$ and $k = 1$.

## M    Visualization of the loss surface

In this section we present a visualization of the loss surface. We adapted the code from [15] to analyse the shape of the loss surface at the end of training for different methods. [15] reported that after adversarial training CO, the loss surface would become non-linear. In particular, they found that the FGSM perturbation seems to be misguided by local maxima very close to the clean image that result in ineffective attacks. We note this was already reported by [32] which proposed to perform a

random step to *escape* those maxima. We argue that adding noise to the random step, when properly implemented, actually prevents those maxima to appear in the first place.

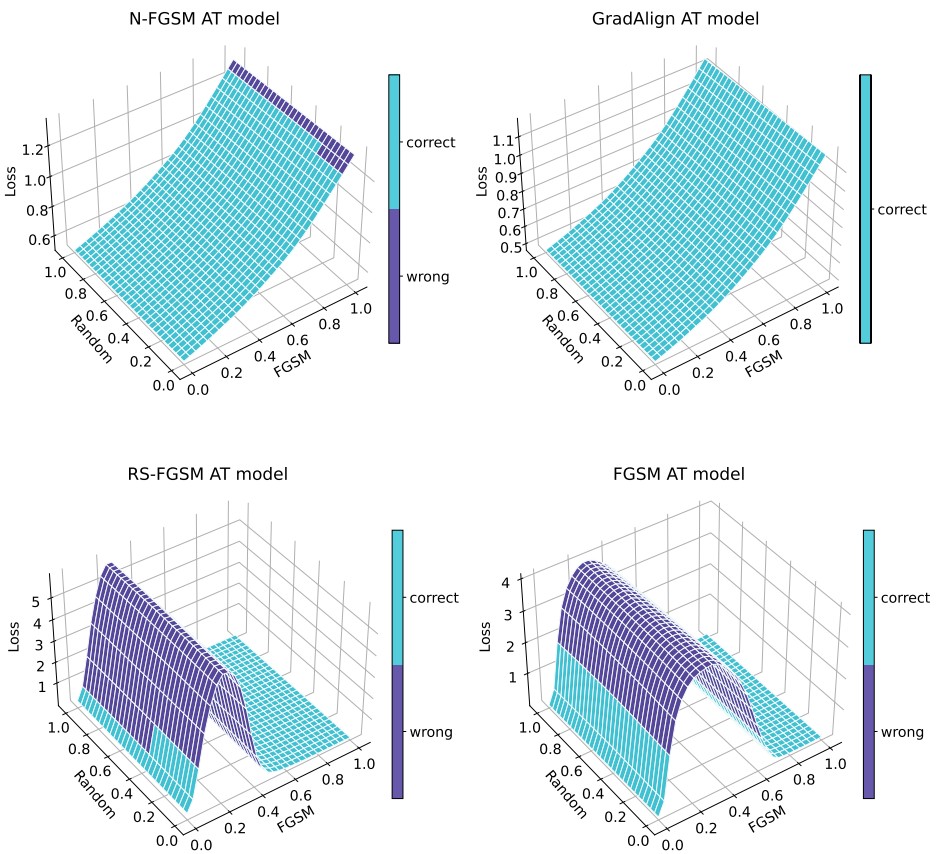

Figure 14: Visualization of the loss surface for models trained using different methods. Given a clean sample from the test set in coordinate $(0, 0)$, we compute the FGSM perturbation and evaluate the loss on the subspace generated by the FGSM perturbation direction and a random direction. That is, we evaluate $x_{\text{clean}} + t_1 \cdot \delta_{\text{FGSM}} + t_2 \cdot \delta_{\text{random}}$, where $t_1, t_2 \in [0, 1]$. Note that FGSM and RS-FGSM both have CO and the final models present a highly non-linear loss surface, on the other hand, both N-FGSM and GradAlign produce final models with a very linear loss surface which is key to obtain meaningful perturbations.

# N   Magnitude of N-FGSM perturbations

**Lemma N.1** (Expected perturbation). *Consider the N-FGSM perturbation as defined in Equation* (3)

$$\delta_{\text{N-FGSM}} = \eta + \alpha \cdot sign\left(\nabla_x \ell(f(x+\eta), y)\right), \;\; where \;\; \eta \sim \Omega.$$

*Let the distribution $\Omega$ be the uniform distribution $\mathcal{U}\left([-k\epsilon, k\epsilon]^d\right)$ and $\alpha > 0$. Then,*

$$\mathbb{E}_\eta \left[\|\delta_{\text{N-FGSM}}\|_2^2\right] = d\left(\frac{k^2\epsilon^2}{3} + \alpha^2\right) \quad and \quad \mathbb{E}_\eta\left[\|\delta_{\text{N-FGSM}}\|_2\right] \leq \sqrt{d\left(\frac{k^2\epsilon^2}{3} + \alpha^2\right)}$$

*Proof.* By Jensen's inequality, we have

$$\mathbb{E}_\eta\left[\|\delta_{\text{N-FGSM}}\|_2\right] \leq \sqrt{\mathbb{E}_\eta\left[\|\delta_{\text{N-FGSM}}\|_2^2\right]}$$

Then let us consider the term $\mathbb{E}_\eta\left[\|\delta_{\text{N-FGSM}}\|_2^2\right]$ and use the shorthand $\nabla(\eta)_i = \left(\nabla_x \ell(f(x+\eta), y)\right)_i$.

$$
\begin{aligned}
\mathbb{E}_\eta\left[\|\delta_{\text{N-FGSM}}\|_2^2\right] =& \mathbb{E}_\eta \|\eta + \alpha \cdot \text{sign}\left(\nabla_x \ell(f(x+\eta), y)\right)\|_2^2 \\
=& \mathbb{E}_\eta\left[\sum_{i=1}^d \left(\eta_i + \alpha \cdot \text{sign}(\nabla(\eta)_i)\right)^2\right] \\
=& \sum_{i=1}^d \mathbb{E}_\eta\left[\left(\eta_i + \alpha \cdot \text{sign}(\nabla(\eta)_i)\right)^2\right] \\
=& \sum_{i=1}^d \mathbb{E}_\eta\left[\left(\eta_i + \alpha \cdot \text{sign}(\nabla(\eta)_i)\right)^2 | \text{sign}(\nabla(\eta)_i) = 1\right] \mathbb{P}_\eta\left[\text{sign}(\nabla(\eta)_i) = 1\right] \\
&+ \sum_{i=1}^d \mathbb{E}_\eta\left[\left(\eta_i + \alpha \cdot \text{sign}(\nabla(\eta)_i)\right)^2 | \text{sign}(\nabla(\eta)_i) = -1\right] \mathbb{P}_\eta\left[\text{sign}(\nabla(\eta)_i) = -1\right] \\
=& \sum_{i=1}^d \frac{1}{2k\epsilon} \int_{-k\epsilon}^{k\epsilon} (\eta_i + \alpha)^2 \, d\eta_i \cdot \mathbb{P}_\eta\left[\text{sign}(\nabla(\eta)_i) = 1\right] \\
&+ \frac{1}{2k\epsilon}\sum_{i=1}^d \int_{-k\epsilon}^{k\epsilon} (\eta_i - \alpha)^2 \, d\eta_i \cdot \mathbb{P}_\eta\left[\text{sign}(\nabla(\eta)_i) = -1\right] \\
=& \sum_{i=1}^d \frac{1}{2k\epsilon} \int_{\alpha-k\epsilon}^{\alpha+k\epsilon} z^2 dz \cdot \mathbb{P}_\eta\left[\text{sign}(\nabla(\eta)_i) = 1\right] \\
&+ \frac{1}{2k\epsilon}\sum_{i=1}^d \int_{-\alpha-k\epsilon}^{-\alpha+k\epsilon} z^2 dz \cdot \mathbb{P}_\eta\left[\text{sign}(\nabla(\eta)_i) = -1\right] \\
=& \sum_{i=1}^d \frac{1}{2k\epsilon} \int_{\alpha-k\epsilon}^{\alpha+k\epsilon} z^2 dz \cdot \mathbb{P}_\eta\left[\text{sign}(\nabla(\eta)_i) = 1\right] \\
&+ \frac{1}{2k\epsilon}\sum_{i=1}^d \int_{\alpha-k\epsilon}^{\alpha+k\epsilon} z^2 dz \cdot \mathbb{P}_\eta\left[\text{sign}(\nabla(\eta)_i) = -1\right] \\
=& \frac{1}{2k\epsilon} \int_{\alpha-k\epsilon}^{\alpha+k\epsilon} z^2 dz \sum_{i=1}^d \left(\mathbb{P}_\eta\left[\text{sign}(\nabla(\eta)_i) = 1\right] + \mathbb{P}_\eta\left[\text{sign}(\nabla(\eta)_i) = -1\right]\right) \\
=& \frac{d}{6k\epsilon}\left[(\alpha + k\epsilon)^3 - (\alpha - k\epsilon)^3\right] = \frac{dk^2\epsilon^2}{3} + d\alpha^2
\end{aligned}
$$

Therefore,

$$\mathbb{E}_\eta \left[ \|\delta_{\text{N-FGSM}}\|_2 \right] \leq \sqrt{d \left( \frac{k^2 \epsilon^2}{3} + \alpha^2 \right)}.$$

□

**Theorem N.2.** *Let $\delta_{N\text{-}FGSM}$ be our proposed single-step method defined by Equation (3), $\delta_{FGSM}$ be the FGSM method [11] and $\delta_{RS\text{-}FGSM}$ be the RS-FGSM method [35]. Then, with default hyperparameter values and for any $\epsilon > 0$, we have that*

$$\mathbb{E}_\eta \left[ \|\delta_{N\text{-}FGSM}\|_2^2 \right] > \mathbb{E}_\eta \left[ \|\delta_{FGSM}\|_2^2 \right] > \mathbb{E}_\eta \left[ \|\delta_{RS\text{-}FGSM}\|_2^2 \right].$$

*Proof.* From Lemma N.1 we have that

$$\mathbb{E}_\eta \left[ \|\delta_{\text{N-FGSM}}\|_2^2 \right] = d \left( \frac{k^2 \epsilon^2}{3} + \alpha^2 \right).$$

On the other hand, [1] showed that

$$\mathbb{E}_\eta \left[ \|\delta_{\text{RS-FGSM}}\|_2^2 \right] = d \left( -\frac{1}{6\epsilon} \alpha^3 + \frac{1}{2} \alpha^2 + \frac{1}{3} \epsilon^2 \right).$$

Finally, we note that

$$\mathbb{E}_\eta \left[ \|\delta_{\text{FGSM}}\|_2^2 \right] = \|\delta_{\text{FGSM}}\|_2^2 = d\epsilon^2.$$

The default hyperparameters for N-FGSM are $k = 2, \ \alpha = \epsilon$ and RS-FGSM uses $\alpha = 5\epsilon/4$. With these hyperparameters and any $\epsilon > 0$ we have

$$\mathbb{E}_\eta \left[ \|\delta_{\text{N-FGSM}}\|_2^2 \right] = \frac{7}{3} d\epsilon^2 > \mathbb{E}_\eta \left[ \|\delta_{\text{FGSM}}\|_2^2 \right] = d\epsilon^2 > \mathbb{E}_\eta \left[ \|\delta_{\text{RS-FGSM}}\|_2^2 \right] = \frac{101}{128} d\epsilon^2$$

□

In Lemma N.1 we compute the expected value of the squared $\ell_2$ norm of N-FGSM perturbations and by Jensen's inequality we obtain an upper bound for the expected $\ell_2$ norm of N-FGSM perturbations. However, obtaining the exact expected magnitude is more complex. To compliment our analytic results, we approximate the $\ell_2$ norm of FGSM, RS-FGSM and N-FGSM via Monte Carlo sampling. Results are presented in Figure 15. We observe that the empirical estimations are very close to the analytical upper bounds and that indeed, N-FGSM has a magnitude significantly above that of FGSM or RS-FGSM.

## O   N-FGSM with Gaussian noise

In the main paper we have only explored noise sources coming from a Uniform distribution. Since we are measuring robustness against $l_\infty-$ attacks, the Uniform distribution is a natural choice because the random perturbations will be bounded to the $l_\infty$ ball defined by the span of the distribution. However, for the sake of completeness, we also explore the performance of augmenting the samples from a Gaussian distribution where we choose its standard deviation to match that of the uniform distribution. In Table 7 we present a comparison of the clean (top) and PGD-50-10 (bottom) accuracy for different values of $\alpha$ and noise magnitude with $\epsilon = 8/255$. Recall that by default we use Uniform distribution $\mathcal{U}[-k, k]$, therefore hyperparameter $k$ sets the noise magnitude.

Increasing the FGSM step size without increasing the amount of noise leads to CO. Note results for $k = 0.5\epsilon$. More importantly, results are very similar when the two noise distributions share the same standard deviation. Thus, using Gaussian instead of Uniform noise does not seem to alter the results. Although this might be expected, we remark that the Gaussian is an unbounded noise distribution and the common practice in adversarial training is to always restrict the norm of the perturbations.

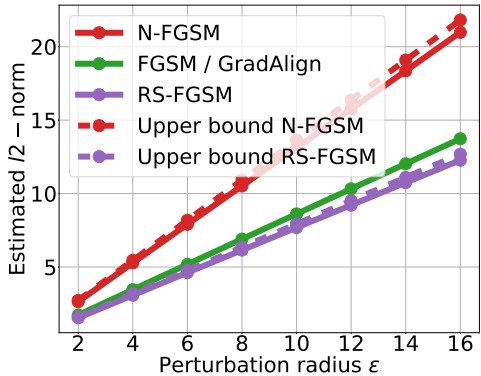

Figure 15: Monte Carlo estimations of the expected $l_2-$norm of perturbations from different methods and corresponding analytical upper bounds. As mentioned in [1], we observe that RS-FGSM perturbations have lower $l_2$ norm than FGSM. However, N-FGSM perturbations have a significantly higher $l_2-$norm than both RS-FGSM and FGSM. This seems to indicate that the role of random step is not simply to lower the $l_2$ norm as previously suggested [1].

Table 7: Comparison of the clean (top) and PGD-50-10 (bottom) accuracy across different values of step-size $\alpha$ and noise magnitude for the Uniform and Gaussian distributions with $\epsilon = 8/255$. For every value of $k$, we use a Gaussian with matching standard deviation. We observe that when we match the standard deviation, both distribution perform similarly.

| | Uniform Noise | | | Gaussian Noise | | |
|---|---|---|---|---|---|---|
| | $\alpha = {}^6/_{255}\ (0.75\epsilon)$ | $\alpha = {}^8/_{255}\ (1\epsilon)$ | $\alpha = {}^{10}/_{255}\ (1.25\epsilon)$ | $\alpha = {}^6/_{255}\ (0.75\epsilon)$ | $\alpha = {}^8/_{255}\ (1\epsilon)$ | $\alpha = {}^{10}/_{255}\ (1.25\epsilon)$ |
| $k = 0.5\epsilon$ | $85.52 \pm 0.23$ | $81.54 \pm 0.19$ | $82.81 \pm 1.11$ | $85.27 \pm 0.11$ | $81.71 \pm 0.27$ | $83.34 \pm 1.48$ |
| | $44.14 \pm 0.24$ | $47.93 \pm 0.11$ | $0.0 \pm 0.0$ | $44.23 \pm 0.17$ | $47.98 \pm 0.14$ | $0.0 \pm 0.0$ |
| $k = 1\epsilon$ | $85.03 \pm 0.09$ | $81.57 \pm 0.07$ | $77.32 \pm 0.14$ | $85.01 \pm 0.17$ | $81.35 \pm 0.14$ | $77.22 \pm 0.32$ |
| | $44.44 \pm 0.13$ | $48.16 \pm 0.21$ | $49.68 \pm 0.25$ | $44.41 \pm 0.04$ | $48.21 \pm 0.11$ | $49.83 \pm 0.1$ |
| $k = 2\epsilon$ | $84.49 \pm 0.1$ | $80.58 \pm 0.22$ | $76.49 \pm 0.14$ | $84.35 \pm 0.24$ | $80.44 \pm 0.31$ | $76.33 \pm 0.37$ |
| | $44.44 \pm 0.15$ | $48.12 \pm 0.07$ | $49.77 \pm 0.37$ | $44.59 \pm 0.22$ | $48.34 \pm 0.1$ | $49.77 \pm 0.23$ |

## P   Training with noise augmented samples

Gilmer et al. [9] and Fawzi et al. [8] report a close link between robustness to adversarial attacks and robustness to random noise. Actually, [9] report that training with noise-augmented samples can improve adversarial accuracy and vice-versa. We note that N-FGSM can actually be seen as a combination of noise-augmentation and adversarial attacks. Here we perform an ablation where we train models with samples augmented with uniform noise $\mathcal{U}[-k, k]$ and then test the PGD-50-10 accuracy. We observe, that indeed random noise can increase the robustness to worst-case perturbations for small $\epsilon - l_\infty$ balls. However, as we increase $\epsilon$, noise augmentation is no longer very effective. With N-FGSM, we apply a weak attack to these noise-augmented samples and this seems to be enough to make them effective for adversarial training.

## Q   Comparison of adversarial training cost

In this section we describe how we compute the relative training cost for single-step methods shown in Figure 1 (right). We approximate the cost based on the number of forward/backward passes each method uses, disregarding the cost of other additional operations such as adding a random step for RS-FGSM or N-FGSM. We understand these operations have a negligible cost compared to a full forward or backward pass.

**FGSM:** FGSM is the cheapest of all methods since it only uses one forward/backward to compute the attack and an additional forward/backward to compute the weight update. Hence, Cost FGSM = 2 F/B.

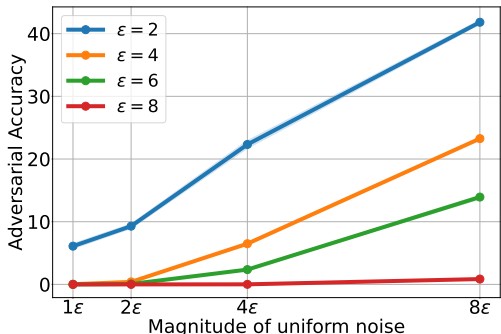

Figure 16: Training with uniform noise augmented samples improves adversarial accuracy for small perturbations but is not effective to protect against larger $l_\infty$ radius $\epsilon$. This motivates us to further augment the noisy samples with FGSM. All experiments are averaged over 3 runs.

**RS-FGSM:** As previously mentioned, we do not take into account the cost of random steps or clipping, hence we consider RS-FGSM to have the same cost as standard FGSM. Cost RS-FGSM = 2 F/B.

**N-FGSM:** Idem as before, cost of N-FGSM = 2 F/B.

**ZeroGrad:** For ZeroGrad they need to do an additional sorting operation to find the smallest gradient components. This could be potentially expensive, however, since the size of the input image is several orders of magnitude smaller than that of the network, we also ignore this cost. Cost ZeroGrad = 2 F/B.

**MultiGrad:** MultiGrad computes 3 random steps and evaluates the gradient in all of them. Therefore, it needs to do 3 F/B to compute the attack and an additional one to update the weights. Cost MultiGrad = 4 F/B.

**[15]:** [15] compute the RS-FGSM perturbation and evaluate the model on $c$ points along this direction. Therefore, they will spend 1F/B on the RS-FGSM attack, $c - 1$ F on the evaluations since the clean image has already been evaluated; and 1 F/B for the weight update. In our plot, we used $c = 3$ since it was the most chosen setting. [15] assume the cost of a forward is similar to that of a backward pass, following this assumption, cost of [15] is 1 F/B + 2 F + 1 F/B = 3 F/B

**Free-AT:** [26] re-use the gradient from the previous backward pass to compute the FGSM perturbation of the current iteration. Hence, the cost of their training is only 1 F/B per iteration. However, [35] observed they needed a longer training schedule to produce comparable results. Therefore, the total training cost per iteration (1 F/B) is scaled by 96 in the case of Free-AT, while it is only scaled by 30 for other methods. Relative cost Free = (96 · 1 F/B) / (30 · 2 F/B).

**GradAlign:** Finally, GradAlign uses FGSM with a regularizer. However, this regularizer needs to compute second-order derivatives via double backpropagation, which does not have the same cost as regular backpropagation. [1] report that the cost of using GradAlign regularizer increased the cost of FGSM by 3.

# R    Infrastructure details and GPU hours

All our training runs have been conducted on either NVIDIA GPU V-100 or P-100 from an internal cluster. The total compute for the results presented in this work is roughly 2500 hours.

# S    Detailed results for Section 5.1 and Section 7

In this section we present the tables with the exact numbers used in plots comparing adversarial training methods. For each method and $\epsilon - l_\infty$ radius, the top number is the clean accuracy while the bottom number is the PGD-50-10 accuracy. We separate single-step from multi-step methods with a double line.

**PreActResNet18 – CIFAR-10 Dataset**

| | $\epsilon = {}^{2}/_{255}$ | $\epsilon = {}^{4}/_{255}$ | $\epsilon = {}^{6}/_{255}$ | $\epsilon = {}^{8}/_{255}$ | $\epsilon = {}^{10}/_{255}$ | $\epsilon = {}^{12}/_{255}$ | $\epsilon = {}^{14}/_{255}$ | $\epsilon = {}^{16}/_{255}$ |
|---|---|---|---|---|---|---|---|---|
| **N-FGSM** | $91.48 \pm 0.17$ | $88.44 \pm 0.09$ | $84.72 \pm 0.04$ | $80.58 \pm 0.22$ | $75.98 \pm 0.1$ | $71.46 \pm 0.14$ | $67.11 \pm 0.37$ | $63.18 \pm 0.49$ |
| | $\mathbf{79.43 \pm 0.21}$ | $\mathbf{67.09 \pm 0.31}$ | $\mathbf{56.62 \pm 0.26}$ | $\mathbf{48.12 \pm 0.07}$ | $\mathbf{41.56 \pm 0.16}$ | $\mathbf{36.43 \pm 0.16}$ | $\mathbf{32.11 \pm 0.2}$ | $\mathbf{27.67 \pm 0.93}$ |
| Grad Align | $91.73 \pm 0.04$ | $88.76 \pm 0.0$ | $85.67 \pm 0.02$ | $81.9 \pm 0.22$ | $77.54 \pm 0.06$ | $73.29 \pm 0.23$ | $68.01 \pm 0.32$ | $61.3 \pm 0.15$ |
| | $79.16 \pm 0.03$ | $\mathbf{67.13 \pm 0.26}$ | $\mathbf{56.27 \pm 0.31}$ | $\mathbf{48.14 \pm 0.15}$ | $40.75 \pm 0.28$ | $34.51 \pm 0.63$ | $30.36 \pm 0.27$ | $\mathbf{26.64 \pm 0.27}$ |
| FGSM | $91.6 \pm 0.1$ | $88.77 \pm 0.04$ | $85.58 \pm 0.11$ | $86.41 \pm 0.7$ | $82.08 \pm 1.62$ | $80.6 \pm 2.59$ | $76.04 \pm 2.37$ | $77.14 \pm 2.46$ |
| | $79.35 \pm 0.06$ | $67.11 \pm 0.09$ | $56.33 \pm 0.41$ | $0.0 \pm 0.0$ | $0.0 \pm 0.0$ | $0.0 \pm 0.0$ | $0.0 \pm 0.0$ | $0.0 \pm 0.0$ |
| RS-FGSM | $92.09 \pm 0.05$ | $89.69 \pm 0.01$ | $87.0 \pm 0.12$ | $84.05 \pm 0.13$ | $85.21 \pm 0.51$ | $65.22 \pm 23.23$ | $43.59 \pm 25.01$ | $76.66 \pm 0.38$ |
| | $78.64 \pm 0.08$ | $66.12 \pm 0.22$ | $54.87 \pm 0.22$ | $46.08 \pm 0.18$ | $0.0 \pm 0.0$ | $0.0 \pm 0.0$ | $0.0 \pm 0.0$ | $0.0 \pm 0.0$ |
| Kim et. al. | $92.85 \pm 0.11$ | $91.1 \pm 0.04$ | $89.34 \pm 0.05$ | $89.02 \pm 0.1$ | $88.27 \pm 0.14$ | $88.35 \pm 0.31$ | $90.01 \pm 0.25$ | $90.45 \pm 0.08$ |
| | $74.74 \pm 0.35$ | $60.51 \pm 0.4$ | $48.95 \pm 0.45$ | $33.01 \pm 0.09$ | $24.43 \pm 0.84$ | $13.11 \pm 0.63$ | $5.86 \pm 0.57$ | $1.88 \pm 0.05$ |
| AT Free | $87.99 \pm 0.16$ | $84.98 \pm 0.13$ | $81.77 \pm 0.11$ | $78.41 \pm 0.18$ | $74.79 \pm 0.22$ | $73.91 \pm 4.19$ | $61.92 \pm 14.94$ | $71.64 \pm 3.89$ |
| | $74.27 \pm 0.33$ | $62.47 \pm 0.25$ | $53.18 \pm 0.15$ | $46.03 \pm 0.36$ | $39.87 \pm 0.07$ | $22.99 \pm 16.26$ | $0.0 \pm 0.0$ | $0.0 \pm 0.0$ |
| ZeroGrad | $91.71 \pm 0.08$ | $88.8 \pm 0.11$ | $85.71 \pm 0.1$ | $82.62 \pm 0.05$ | $79.91 \pm 0.12$ | $78.11 \pm 0.2$ | $75.66 \pm 0.46$ | $75.42 \pm 0.13$ |
| | $79.36 \pm 0.05$ | $\mathbf{67.32 \pm 0.02}$ | $56.14 \pm 0.21$ | $47.08 \pm 0.1$ | $37.58 \pm 0.2$ | $27.41 \pm 0.27$ | $21.29 \pm 0.97$ | $13.06 \pm 0.22$ |
| MultiGrad | $91.57 \pm 0.16$ | $88.74 \pm 0.12$ | $85.75 \pm 0.05$ | $82.33 \pm 0.14$ | $78.73 \pm 0.16$ | $75.28 \pm 0.2$ | $80.94 \pm 5.94$ | $71.42 \pm 5.63$ |
| | $79.34 \pm 0.02$ | $66.81 \pm 0.02$ | $56.02 \pm 0.3$ | $47.29 \pm 0.07$ | $40.11 \pm 0.24$ | $33.87 \pm 0.17$ | $9.55 \pm 13.5$ | $16.35 \pm 11.57$ |
| PGD-2 | $91.4 \pm 0.07$ | $88.46 \pm 0.13$ | $85.14 \pm 0.13$ | $81.41 \pm 0.05$ | $77.18 \pm 0.15$ | $72.9 \pm 0.26$ | $70.39 \pm 2.71$ | $64.81 \pm 11.58$ |
| | $\mathbf{79.55 \pm 0.15}$ | $67.62 \pm 0.03$ | $57.39 \pm 0.13$ | $49.58 \pm 0.08$ | $43.3 \pm 0.11$ | $38.13 \pm 0.15$ | $22.89 \pm 15.26$ | $9.6 \pm 13.37$ |
| PGD-10 | $91.25 \pm 0.04$ | $88.34 \pm 0.11$ | $84.79 \pm 0.11$ | $80.71 \pm 0.14$ | $76.13 \pm 0.35$ | $71.24 \pm 0.3$ | $66.7 \pm 0.39$ | $62.11 \pm 0.62$ |
| | $\mathbf{79.47 \pm 0.13}$ | $\mathbf{68.29 \pm 0.24}$ | $\mathbf{58.85 \pm 0.18}$ | $\mathbf{51.33 \pm 0.31}$ | $\mathbf{45.02 \pm 0.49}$ | $\mathbf{39.93 \pm 0.5}$ | $\mathbf{36.02 \pm 0.67}$ | $\mathbf{32.22 \pm 0.64}$ |

**PreActResNet18 – CIFAR-100 Dataset**

| | $\epsilon = {}^{2}/_{255}$ | $\epsilon = {}^{4}/_{255}$ | $\epsilon = {}^{6}/_{255}$ | $\epsilon = {}^{8}/_{255}$ | $\epsilon = {}^{10}/_{255}$ | $\epsilon = {}^{12}/_{255}$ | $\epsilon = {}^{14}/_{255}$ | $\epsilon = {}^{16}/_{255}$ |
|---|---|---|---|---|---|---|---|---|
| **N-FGSM** | $69.12 \pm 0.27$ | $64.0 \pm 0.06$ | $59.53 \pm 0.02$ | $54.9 \pm 0.2$ | $50.6 \pm 0.16$ | $46.06 \pm 0.14$ | $41.67 \pm 0.25$ | $37.91 \pm 0.11$ |
| | $\mathbf{51.02 \pm 0.34}$ | $\mathbf{39.5 \pm 0.12}$ | $\mathbf{32.06 \pm 0.37}$ | $\mathbf{26.46 \pm 0.22}$ | $\mathbf{22.23 \pm 0.17}$ | $\mathbf{18.95 \pm 0.15}$ | $\mathbf{16.33 \pm 0.15}$ | $\mathbf{14.34 \pm 0.07}$ |
| Grad Align | $68.96 \pm 0.15$ | $64.71 \pm 0.16$ | $60.42 \pm 0.23$ | $56.53 \pm 0.31$ | $54.06 \pm 0.44$ | $48.87 \pm 0.32$ | $43.84 \pm 0.14$ | $38.93 \pm 0.21$ |
| | $\mathbf{51.31 \pm 0.12}$ | $\mathbf{39.37 \pm 0.25}$ | $\mathbf{31.91 \pm 0.28}$ | $25.8 \pm 0.14$ | $18.7 \pm 1.92$ | $17.86 \pm 0.04$ | $15.51 \pm 0.16$ | $13.62 \pm 0.19$ |
| FGSM | $69.01 \pm 0.13$ | $64.47 \pm 0.15$ | $63.85 \pm 2.18$ | $53.42 \pm 0.65$ | $45.06 \pm 2.29$ | $46.14 \pm 2.58$ | $41.66 \pm 0.88$ | $44.68 \pm 1.74$ |
| | $51.3 \pm 0.19$ | $39.7 \pm 0.16$ | $10.93 \pm 14.64$ | $0.0 \pm 0.0$ | $0.0 \pm 0.0$ | $0.0 \pm 0.0$ | $0.0 \pm 0.0$ | $0.0 \pm 0.0$ |
| RS-FGSM | $69.83 \pm 0.29$ | $65.9 \pm 0.36$ | $62.15 \pm 0.23$ | $55.26 \pm 6.86$ | $32.33 \pm 12.12$ | $36.07 \pm 2.59$ | $21.52 \pm 5.56$ | $20.38 \pm 6.15$ |
| | $50.13 \pm 0.32$ | $38.36 \pm 0.19$ | $30.82 \pm 0.08$ | $0.01 \pm 0.01$ | $0.0 \pm 0.0$ | $0.0 \pm 0.0$ | $0.0 \pm 0.0$ | $0.0 \pm 0.0$ |
| Kim et. al. | $72.92 \pm 0.41$ | $70.16 \pm 0.07$ | $67.98 \pm 0.19$ | $68.07 \pm 0.1$ | $68.37 \pm 0.21$ | $74.09 \pm 0.06$ | $74.06 \pm 0.34$ | $74.01 \pm 0.36$ |
| | $44.19 \pm 0.25$ | $30.63 \pm 0.28$ | $22.0 \pm 0.02$ | $12.75 \pm 0.21$ | $6.98 \pm 0.23$ | $0.0 \pm 0.0$ | $0.0 \pm 0.0$ | $0.0 \pm 0.0$ |
| AT Free | $63.01 \pm 0.19$ | $59.41 \pm 0.27$ | $55.43 \pm 0.37$ | $51.91 \pm 0.08$ | $48.11 \pm 0.09$ | $43.48 \pm 1.25$ | $18.33 \pm 4.86$ | $20.43 \pm 11.25$ |
| | $45.7 \pm 0.33$ | $35.95 \pm 0.09$ | $29.37 \pm 0.21$ | $24.32 \pm 0.4$ | $20.64 \pm 0.22$ | $5.71 \pm 8.05$ | $0.0 \pm 0.0$ | $0.0 \pm 0.0$ |
| ZeroGrad | $69.35 \pm 0.36$ | $64.59 \pm 0.32$ | $60.69 \pm 0.09$ | $56.94 \pm 0.13$ | $54.55 \pm 0.17$ | $52.97 \pm 0.34$ | $50.87 \pm 0.26$ | $50.73 \pm 0.3$ |
| | $\mathbf{51.1 \pm 0.09}$ | $\mathbf{39.38 \pm 0.15}$ | $31.72 \pm 0.21$ | $25.87 \pm 0.09$ | $19.49 \pm 0.08$ | $14.32 \pm 0.08$ | $10.92 \pm 0.59$ | $7.3 \pm 0.16$ |
| MultiGrad | $69.01 \pm 0.16$ | $64.44 \pm 0.11$ | $60.65 \pm 0.26$ | $56.84 \pm 0.2$ | $53.62 \pm 0.25$ | $53.05 \pm 1.85$ | $48.28 \pm 0.66$ | $45.28 \pm 11.14$ |
| | $51.15 \pm 0.03$ | $39.16 \pm 0.03$ | $31.73 \pm 0.09$ | $25.96 \pm 0.11$ | $21.37 \pm 0.16$ | $9.57 \pm 7.32$ | $3.2 \pm 4.49$ | $0.0 \pm 0.0$ |
| PGD-2 | $69.18 \pm 0.1$ | $64.32 \pm 0.14$ | $60.21 \pm 0.13$ | $55.8 \pm 0.16$ | $51.68 \pm 0.1$ | $48.2 \pm 0.1$ | $46.14 \pm 1.24$ | $37.97 \pm 10.52$ |
| | $\mathbf{51.36 \pm 0.03}$ | $40.06 \pm 0.14$ | $32.99 \pm 0.24$ | $27.38 \pm 0.16$ | $23.39 \pm 0.19$ | $19.83 \pm 0.29$ | $10.55 \pm 7.51$ | $4.79 \pm 6.75$ |
| PGD-10 | $68.83 \pm 0.07$ | $63.87 \pm 0.09$ | $59.37 \pm 0.07$ | $54.79 \pm 0.38$ | $50.53 \pm 0.15$ | $46.05 \pm 0.21$ | $41.76 \pm 0.07$ | $37.81 \pm 0.14$ |
| | $\mathbf{51.51 \pm 0.27}$ | $\mathbf{40.59 \pm 0.36}$ | $\mathbf{33.65 \pm 0.02}$ | $\mathbf{28.55 \pm 0.27}$ | $\mathbf{24.17 \pm 0.12}$ | $\mathbf{21.2 \pm 0.12}$ | $\mathbf{18.72 \pm 0.06}$ | $\mathbf{16.59 \pm 0.16}$ |

**PreActResNet18 – SVHN Dataset**

| | $\epsilon = 2/255$ | $\epsilon = 4/255$ | $\epsilon = 6/255$ | $\epsilon = 8/255$ | $\epsilon = 10/255$ | $\epsilon = 12/255$ |
|---|---|---|---|---|---|---|
| **N-FGSM** | $96.01 \pm 0.04$ | $94.54 \pm 0.15$ | $92.25 \pm 0.33$ | $89.56 \pm 0.49$ | $86.74 \pm 0.86$ | $81.48 \pm 1.64$ |
| | $\mathbf{86.44 \pm 0.1}$ | $\mathbf{72.53 \pm 0.19}$ | $58.42 \pm 0.14$ | $\mathbf{45.63 \pm 0.11}$ | $\mathbf{33.96 \pm 0.49}$ | $\mathbf{26.13 \pm 0.81}$ |
| Grad Align | $96.02 \pm 0.05$ | $94.56 \pm 0.21$ | $92.53 \pm 0.24$ | $90.1 \pm 0.34$ | $87.23 \pm 0.75$ | $84.01 \pm 0.46$ |
| | $\mathbf{86.43 \pm 0.1}$ | $72.12 \pm 0.19$ | $57.34 \pm 0.24$ | $43.85 \pm 0.14$ | $32.87 \pm 0.33$ | $23.62 \pm 0.41$ |
| FGSM | $96.04 \pm 0.07$ | $95.67 \pm 0.07$ | $93.73 \pm 0.68$ | $91.74 \pm 0.86$ | $90.76 \pm 0.63$ | $87.17 \pm 0.43$ |
| | $\mathbf{86.5 \pm 0.05}$ | $13.61 \pm 5.83$ | $0.56 \pm 0.72$ | $0.26 \pm 0.36$ | $0.07 \pm 0.1$ | $0.0 \pm 0.0$ |
| RS-FGSM | $96.18 \pm 0.11$ | $95.09 \pm 0.09$ | $95.11 \pm 0.44$ | $94.46 \pm 0.16$ | $93.88 \pm 0.24$ | $92.74 \pm 0.5$ |
| | $86.16 \pm 0.14$ | $71.28 \pm 0.4$ | $0.11 \pm 0.08$ | $0.0 \pm 0.0$ | $0.0 \pm 0.0$ | $0.0 \pm 0.0$ |
| Kim et. al. | $96.35 \pm 0.02$ | $95.25 \pm 0.08$ | $94.83 \pm 0.02$ | $94.88 \pm 0.29$ | $96.61 \pm 0.09$ | $96.61 \pm 0.01$ |
| | $83.26 \pm 0.24$ | $66.32 \pm 0.63$ | $48.27 \pm 0.52$ | $31.8 \pm 1.1$ | $0.18 \pm 0.21$ | $0.0 \pm 0.0$ |
| AT Free | $95.01 \pm 0.09$ | $93.66 \pm 0.12$ | $91.72 \pm 0.29$ | $91.29 \pm 4.07$ | $91.86 \pm 3.66$ | $92.36 \pm 1.0$ |
| | $84.55 \pm 0.27$ | $71.61 \pm 0.75$ | $\mathbf{59.31 \pm 1.0}$ | $0.01 \pm 0.0$ | $0.0 \pm 0.0$ | $0.0 \pm 0.0$ |
| ZeroGrad | $96.06 \pm 0.03$ | $94.81 \pm 0.16$ | $93.53 \pm 0.26$ | $92.42 \pm 1.29$ | $90.34 \pm 0.32$ | $88.09 \pm 0.4$ |
| | $\mathbf{86.43 \pm 0.1}$ | $71.59 \pm 0.22$ | $51.72 \pm 0.53$ | $35.93 \pm 2.73$ | $21.34 \pm 0.31$ | $14.14 \pm 0.32$ |
| MultiGrad | $96.01 \pm 0.08$ | $94.71 \pm 0.17$ | $95.75 \pm 0.58$ | $94.86 \pm 0.97$ | $94.7 \pm 0.12$ | $94.48 \pm 0.19$ |
| | $\mathbf{86.4 \pm 0.08}$ | $\mathbf{71.98 \pm 0.26}$ | $28.1 \pm 18.85$ | $11.49 \pm 16.19$ | $0.0 \pm 0.0$ | $0.0 \pm 0.0$ |
| PGD-2 | $96.03 \pm 0.14$ | $94.66 \pm 0.1$ | $93.77 \pm 0.61$ | $94.63 \pm 1.29$ | $84.09 \pm 14.99$ | $94.16 \pm 0.54$ |
| | $86.72 \pm 0.06$ | $73.29 \pm 0.29$ | $60.53 \pm 0.73$ | $20.68 \pm 18.56$ | $0.41 \pm 0.29$ | $0.02 \pm 0.03$ |
| PGD-10 | $95.92 \pm 0.08$ | $94.37 \pm 0.13$ | $92.46 \pm 0.25$ | $89.67 \pm 0.34$ | $85.75 \pm 0.65$ | $80.08 \pm 0.93$ |
| | $\mathbf{86.94 \pm 0.14}$ | $\mathbf{74.76 \pm 0.19}$ | $\mathbf{63.9 \pm 0.48}$ | $\mathbf{53.95 \pm 0.55}$ | $\mathbf{44.91 \pm 0.45}$ | $\mathbf{37.65 \pm 0.53}$ |

**WideResNet28-10 – CIFAR-10 Dataset**

| | $\epsilon = 2/255$ | $\epsilon = 4/255$ | $\epsilon = 6/255$ | $\epsilon = 8/255$ | $\epsilon = 10/255$ | $\epsilon = 12/255$ | $\epsilon = 14/255$ | $\epsilon = 16/255$ |
|---|---|---|---|---|---|---|---|---|
| **N-FGSM** | $92.51 \pm 0.11$ | $89.65 \pm 0.09$ | $85.8 \pm 0.23$ | $81.59 \pm 0.32$ | $76.92 \pm 0.04$ | $72.13 \pm 0.15$ | $67.82 \pm 0.43$ | $56.73 \pm 0.42$ |
| | $\mathbf{81.43 \pm 0.3}$ | $69.11 \pm 0.24$ | $58.29 \pm 0.14$ | $49.53 \pm 0.25$ | $\mathbf{42.37 \pm 0.36}$ | $\mathbf{36.85 \pm 0.2}$ | $\mathbf{31.66 \pm 0.6}$ | $25.01 \pm 0.23$ |
| Grad Align | $92.59 \pm 0.05$ | $89.95 \pm 0.3$ | $86.98 \pm 0.06$ | $83.19 \pm 0.26$ | $79.35 \pm 0.26$ | $73.79 \pm 0.72$ | $66.38 \pm 0.53$ | $57.75 \pm 0.75$ |
| | $81.33 \pm 0.4$ | $\mathbf{69.81 \pm 0.47}$ | $\mathbf{59.0 \pm 0.13}$ | $\mathbf{50.0 \pm 0.05}$ | $41.48 \pm 0.51$ | $35.06 \pm 0.74$ | $30.83 \pm 0.39$ | $\mathbf{26.26 \pm 0.13}$ |
| FGSM | $92.65 \pm 0.17$ | $90.06 \pm 0.18$ | $87.99 \pm 1.3$ | $86.46 \pm 0.45$ | $82.67 \pm 1.78$ | $80.14 \pm 1.2$ | $74.54 \pm 4.01$ | $71.56 \pm 3.78$ |
| | $\mathbf{81.38 \pm 0.22}$ | $69.59 \pm 0.25$ | $38.69 \pm 26.54$ | $0.0 \pm 0.0$ | $0.0 \pm 0.0$ | $0.0 \pm 0.0$ | $0.0 \pm 0.0$ | $0.0 \pm 0.0$ |
| RS-FGSM | $92.85 \pm 0.1$ | $90.73 \pm 0.2$ | $88.24 \pm 0.19$ | $83.64 \pm 1.74$ | $82.1 \pm 1.45$ | $78.62 \pm 0.7$ | $73.25 \pm 8.16$ | $68.64 \pm 4.3$ |
| | $80.9 \pm 0.13$ | $68.23 \pm 0.17$ | $57.21 \pm 0.17$ | $0.0 \pm 0.0$ | $0.0 \pm 0.0$ | $0.0 \pm 0.0$ | $0.0 \pm 0.0$ | $0.0 \pm 0.0$ |
| RandAlpha | $93.37 \pm 0.22$ | $92.17 \pm 0.21$ | $90.71 \pm 0.14$ | $89.16 \pm 0.19$ | $87.44 \pm 0.31$ | $85.69 \pm 0.28$ | $83.98 \pm 0.24$ | $83.23 \pm 0.46$ |
| | $77.67 \pm 0.66$ | $63.73 \pm 0.31$ | $50.4 \pm 0.14$ | $39.37 \pm 0.42$ | $30.13 \pm 0.9$ | $23.13 \pm 0.33$ | $16.0 \pm 0.22$ | $8.47 \pm 0.66$ |
| AT Free | $90.66 \pm 0.25$ | $88.37 \pm 0.15$ | $86.11 \pm 0.29$ | $83.5 \pm 0.27$ | $80.52 \pm 0.32$ | $83.59 \pm 1.35$ | $39.58 \pm 15.8$ | $42.59 \pm 27.96$ |
| | $77.0 \pm 0.27$ | $64.25 \pm 0.33$ | $53.76 \pm 0.48$ | $44.85 \pm 0.39$ | $31.87 \pm 5.53$ | $0.0 \pm 0.0$ | $0.0 \pm 0.0$ | $0.0 \pm 0.0$ |
| ZeroGrad | $92.62 \pm 0.11$ | $90.17 \pm 0.05$ | $86.98 \pm 0.28$ | $84.25 \pm 0.28$ | $81.72 \pm 0.29$ | $79.24 \pm 0.82$ | $78.14 \pm 0.46$ | $75.34 \pm 0.12$ |
| | $\mathbf{81.42 \pm 0.28}$ | $69.28 \pm 0.29$ | $58.4 \pm 0.14$ | $48.29 \pm 0.16$ | $36.08 \pm 0.29$ | $28.24 \pm 1.79$ | $18.54 \pm 0.31$ | $14.6 \pm 0.12$ |
| MultiGrad | $92.64 \pm 0.1$ | $90.18 \pm 0.13$ | $87.11 \pm 0.36$ | $83.87 \pm 0.46$ | $80.89 \pm 0.14$ | $82.88 \pm 2.85$ | $86.6 \pm 1.52$ | $85.46 \pm 3.73$ |
| | $\mathbf{81.19 \pm 0.28}$ | $69.3 \pm 0.2$ | $57.98 \pm 0.08$ | $48.74 \pm 0.09$ | $41.22 \pm 0.57$ | $4.46 \pm 6.09$ | $0.0 \pm 0.0$ | $0.0 \pm 0.0$ |
| PGD-2 | $92.69 \pm 0.14$ | $90.18 \pm 0.19$ | $86.87 \pm 0.18$ | $83.31 \pm 0.16$ | $79.61 \pm 0.47$ | $75.81 \pm 0.24$ | $71.41 \pm 1.38$ | $67.2 \pm 14.94$ |
| | $\mathbf{81.54 \pm 0.18}$ | $69.87 \pm 0.26$ | $59.4 \pm 0.19$ | $50.88 \pm 0.16$ | $43.94 \pm 0.24$ | $37.77 \pm 0.57$ | $21.06 \pm 13.39$ | $0.0 \pm 0.0$ |
| PGD-10 | $92.24 \pm 0.31$ | $89.65 \pm 0.33$ | $86.91 \pm 0.51$ | $82.82 \pm 0.7$ | $78.63 \pm 0.66$ | $74.0 \pm 0.67$ | $68.6 \pm 0.58$ | $64.17 \pm 0.72$ |
| | $81.18 \pm 0.57$ | $\mathbf{70.34 \pm 0.26}$ | $\mathbf{60.59 \pm 0.21}$ | $\mathbf{52.58 \pm 0.2}$ | $\mathbf{45.92 \pm 0.38}$ | $\mathbf{40.44 \pm 0.17}$ | $\mathbf{35.98 \pm 0.56}$ | $\mathbf{32.5 \pm 0.61}$ |

**WideResNet28-10 – CIFAR-100 Dataset**

| | $\epsilon = {2}/{255}$ | $\epsilon = {5}/{255}$ | $\epsilon = {6}/{255}$ | $\epsilon = {8}/{255}$ | $\epsilon = {10}/{255}$ | $\epsilon = {12}/{255}$ | $\epsilon = {14}/{255}$ | $\epsilon = {16}/{255}$ |
|---|---|---|---|---|---|---|---|---|
| **N-FGSM** | $71.56 \pm 0.13$ | $66.49 \pm 0.46$ | $61.38 \pm 0.68$ | $56.23 \pm 0.59$ | $51.54 \pm 0.63$ | $46.43 \pm 0.61$ | $42.11 \pm 0.32$ | $38.34 \pm 0.47$ |
| | $\mathbf{52.23 \pm 0.33}$ | $\mathbf{39.93 \pm 0.37}$ | $30.97 \pm 0.21$ | $\mathbf{26.77 \pm 0.65}$ | $\mathbf{23.03 \pm 0.54}$ | $\mathbf{19.3 \pm 0.59}$ | $\mathbf{16.67 \pm 0.4}$ | $\mathbf{14.27 \pm 0.33}$ |
| Grad Align | $71.68 \pm 0.33$ | $67.09 \pm 0.19$ | $62.86 \pm 0.1$ | $58.55 \pm 0.41$ | $53.85 \pm 0.73$ | $46.94 \pm 0.86$ | $42.63 \pm 0.5$ | $36.17 \pm 0.45$ |
| | $51.5 \pm 0.45$ | $\mathbf{39.9 \pm 0.42}$ | $\mathbf{32.0 \pm 0.22}$ | $26.9 \pm 0.62$ | $\mathbf{22.63 \pm 0.62}$ | $\mathbf{19.9 \pm 0.65}$ | $\mathbf{16.93 \pm 0.12}$ | $\mathbf{14.03 \pm 0.24}$ |
| FGSM | $71.92 \pm 0.33$ | $67.34 \pm 0.36$ | $64.72 \pm 1.12$ | $56.87 \pm 1.24$ | $52.31 \pm 2.11$ | $48.99 \pm 1.17$ | $44.27 \pm 1.4$ | $42.05 \pm 1.03$ |
| | $\mathbf{52.83 \pm 0.37}$ | $\mathbf{39.83 \pm 0.31}$ | $0.0 \pm 0.0$ | $0.03 \pm 0.05$ | $0.0 \pm 0.0$ | $0.0 \pm 0.0$ | $0.0 \pm 0.0$ | $0.0 \pm 0.0$ |
| RS-FGSM | $72.65 \pm 0.28$ | $68.26 \pm 0.2$ | $65.58 \pm 0.69$ | $54.25 \pm 5.85$ | $46.08 \pm 4.87$ | $35.84 \pm 0.17$ | $24.4 \pm 1.25$ | $21.37 \pm 5.04$ |
| | $51.63 \pm 0.52$ | $39.57 \pm 0.09$ | $26.63 \pm 2.8$ | $0.0 \pm 0.0$ | $0.0 \pm 0.0$ | $0.0 \pm 0.0$ | $0.0 \pm 0.0$ | $0.0 \pm 0.0$ |
| RandAlpha | $73.9 \pm 0.15$ | $71.17 \pm 0.12$ | $68.65 \pm 0.22$ | $66.42 \pm 0.13$ | $64.05 \pm 0.5$ | $61.99 \pm 0.6$ | $59.74 \pm 0.57$ | $58.9 \pm 0.78$ |
| | $49.13 \pm 0.91$ | $34.3 \pm 0.54$ | $25.5 \pm 0.33$ | $20.27 \pm 0.98$ | $16.3 \pm 0.14$ | $12.4 \pm 0.29$ | $6.93 \pm 0.19$ | $3.63 \pm 0.12$ |
| AT Free | $67.62 \pm 0.24$ | $63.27 \pm 0.72$ | $59.53 \pm 0.31$ | $55.77 \pm 0.28$ | $47.02 \pm 3.83$ | $33.52 \pm 9.24$ | $7.87 \pm 1.78$ | $20.92 \pm 21.48$ |
| | $48.07 \pm 0.31$ | $37.93 \pm 0.69$ | $29.7 \pm 0.51$ | $24.43 \pm 0.37$ | $3.23 \pm 4.43$ | $0.0 \pm 0.0$ | $0.0 \pm 0.0$ | $0.0 \pm 0.0$ |
| ZeroGrad | $71.68 \pm 0.07$ | $67.2 \pm 0.14$ | $63.69 \pm 0.14$ | $60.77 \pm 0.26$ | $61.05 \pm 0.38$ | $58.39 \pm 0.16$ | $56.19 \pm 0.11$ | $56.38 \pm 0.18$ |
| | $\mathbf{52.63 \pm 0.61}$ | $39.57 \pm 0.33$ | $30.27 \pm 0.54$ | $23.7 \pm 0.08$ | $15.1 \pm 0.49$ | $11.13 \pm 0.68$ | $8.8 \pm 0.36$ | $4.9 \pm 0.36$ |
| MultiGrad | $71.8 \pm 0.15$ | $67.73 \pm 0.48$ | $63.24 \pm 0.33$ | $60.05 \pm 0.79$ | $56.39 \pm 0.49$ | $56.79 \pm 8.27$ | $59.8 \pm 3.77$ | $52.96 \pm 5.58$ |
| | $51.9 \pm 0.29$ | $39.7 \pm 0.37$ | $31.5 \pm 0.62$ | $26.03 \pm 0.09$ | $20.8 \pm 0.29$ | $0.0 \pm 0.0$ | $0.0 \pm 0.0$ | $0.0 \pm 0.0$ |
| PGD-2 | $71.62 \pm 0.15$ | $67.25 \pm 0.43$ | $63.18 \pm 0.36$ | $59.02 \pm 0.4$ | $54.47 \pm 0.45$ | $50.91 \pm 0.35$ | $41.03 \pm 3.18$ | $40.13 \pm 3.66$ |
| | $51.73 \pm 0.48$ | $\mathbf{40.27 \pm 0.7}$ | $32.23 \pm 0.19$ | $27.13 \pm 0.37$ | $23.43 \pm 0.31$ | $20.23 \pm 0.39$ | $0.03 \pm 0.05$ | $0.0 \pm 0.0$ |
| PGD-10 | $71.11 \pm 0.62$ | $66.9 \pm 0.57$ | $62.05 \pm 0.47$ | $57.64 \pm 0.81$ | $52.84 \pm 0.88$ | $48.14 \pm 0.73$ | $43.14 \pm 0.87$ | $39.2 \pm 0.62$ |
| | $\mathbf{52.5 \pm 0.59}$ | $\mathbf{40.73 \pm 0.56}$ | $\mathbf{32.8 \pm 0.29}$ | $\mathbf{27.97 \pm 0.59}$ | $\mathbf{24.7 \pm 0.36}$ | $\mathbf{21.8 \pm 0.57}$ | $\mathbf{18.87 \pm 0.6}$ | $\mathbf{16.8 \pm 0.57}$ |

**WideResNet28-10 – SVHN Dataset**

| | $\epsilon = {2}/{255}$ | $\epsilon = {4}/{255}$ | $\epsilon = {6}/{255}$ | $\epsilon = {8}/{255}$ | $\epsilon = {10}/{255}$ | $\epsilon = {12}/{255}$ |
|---|---|---|---|---|---|---|
| **N-FGSM** | $95.64 \pm 0.09$ | $93.66 \pm 0.41$ | $91.77 \pm 0.42$ | $88.89 \pm 0.58$ | $88.07 \pm 0.59$ | $87.52 \pm 0.49$ |
| | $\mathbf{84.1 \pm 0.73}$ | $66.9 \pm 0.86$ | $\mathbf{53.0 \pm 0.36}$ | $\mathbf{40.5 \pm 0.37}$ | $\mathbf{30.47 \pm 0.76}$ | $\mathbf{22.43 \pm 0.53}$ |
| Grad Align | $95.41 \pm 0.06$ | $93.9 \pm 0.48$ | $68.36 \pm 34.49$ | $42.62 \pm 32.73$ | $19.3 \pm 0.21$ | $19.53 \pm 0.08$ |
| | $\mathbf{84.57 \pm 0.56}$ | $67.27 \pm 0.54$ | $39.53 \pm 14.89$ | $24.7 \pm 9.34$ | $17.63 \pm 0.62$ | $18.13 \pm 0.52$ |
| FGSM | $95.83 \pm 0.1$ | $95.0 \pm 0.24$ | $94.23 \pm 0.79$ | $91.11 \pm 1.36$ | $88.83 \pm 1.71$ | $86.74 \pm 0.7$ |
| | $\mathbf{85.03 \pm 0.37}$ | $31.53 \pm 6.57$ | $1.7 \pm 1.36$ | $0.13 \pm 0.19$ | $0.0 \pm 0.0$ | $0.0 \pm 0.0$ |
| RS-FGSM | $95.81 \pm 0.25$ | $94.53 \pm 0.4$ | $95.23 \pm 0.26$ | $94.68 \pm 0.62$ | $93.9 \pm 0.52$ | $91.64 \pm 2.98$ |
| | $83.8 \pm 0.43$ | $66.67 \pm 0.65$ | $0.53 \pm 0.26$ | $0.0 \pm 0.0$ | $0.0 \pm 0.0$ | $0.0 \pm 0.0$ |
| RandAlpha | $96.02 \pm 0.23$ | $95.47 \pm 0.18$ | $94.69 \pm 0.26$ | $93.72 \pm 0.44$ | $93.08 \pm 1.45$ | $93.96 \pm 0.68$ |
| | $82.5 \pm 0.45$ | $63.33 \pm 0.53$ | $47.7 \pm 0.99$ | $35.73 \pm 0.34$ | $23.17 \pm 1.97$ | $11.1 \pm 3.05$ |
| AT Free | $94.85 \pm 0.39$ | $92.95 \pm 0.65$ | $91.62 \pm 1.93$ | $93.74 \pm 0.69$ | $92.47 \pm 0.97$ | $90.5 \pm 1.41$ |
| | $83.13 \pm 0.17$ | $\mathbf{68.67 \pm 0.53}$ | $\mathbf{54.93 \pm 2.58}$ | $0.03 \pm 0.05$ | $0.0 \pm 0.0$ | $0.0 \pm 0.0$ |
| ZeroGrad | $95.78 \pm 0.21$ | $94.06 \pm 0.52$ | $92.13 \pm 0.98$ | $91.04 \pm 0.4$ | $88.85 \pm 0.92$ | $89.8 \pm 1.36$ |
| | $\mathbf{84.47 \pm 0.83}$ | $66.1 \pm 0.37$ | $47.3 \pm 0.62$ | $29.33 \pm 0.56$ | $20.77 \pm 0.63$ | $9.33 \pm 0.76$ |
| MultiGrad | $95.63 \pm 0.16$ | $94.27 \pm 0.38$ | $93.64 \pm 1.21$ | $94.83 \pm 1.55$ | $95.26 \pm 0.34$ | $95.22 \pm 0.15$ |
| | $\mathbf{84.37 \pm 0.59}$ | $67.27 \pm 0.31$ | $50.1 \pm 0.9$ | $1.77 \pm 1.72$ | $0.0 \pm 0.0$ | $0.0 \pm 0.0$ |
| PGD-2 | $95.88 \pm 0.35$ | $94.66 \pm 0.1$ | $93.77 \pm 0.61$ | $92.99 \pm 1.11$ | $88.81 \pm 0.93$ | $83.17 \pm 4.78$ |
| | $\mathbf{86.25 \pm 0.7}$ | $73.29 \pm 0.25$ | $60.53 \pm 0.72$ | $40.77 \pm 4.39$ | $34.33 \pm 2.76$ | $26.8 \pm 3.31$ |
| PGD-10 | $95.92 \pm 0.08$ | $94.36 \pm 0.13$ | $92.46 \pm 0.25$ | $89.67 \pm 0.34$ | $85.98 \pm 0.59$ | $80.08 \pm 0.93$ |
| | $\mathbf{86.94 \pm 0.13}$ | $\mathbf{74.46 \pm 0.54}$ | $\mathbf{63.87 \pm 0.49}$ | $\mathbf{53.95 \pm 0.55}$ | $\mathbf{44.59 \pm 0.14}$ | $\mathbf{37.64 \pm 0.49}$ |