# OpenReview forum: "Make Some Noise: Reliable and Efficient Single-Step Adversarial Training"
_NeurIPS.cc/2022/Conference — NeurIPS 2022 Accept_

### Official Review · Reviewer_cQ7D · 2022-07-11

**Rating:** 7
**Confidence:** 4
**Soundness:** 3 good
**Presentation:** 3 good
**Contribution:** 3 good

**Summary:**

The authors have proposed a single-step adversarial training procedure, termed Noise-FGSM (N-FGSM), by considering added noise step as data augmentation rather than part of the attack. The resultant method centers adversarial perturbations with respect to noise-augmented samples (rather than wrt the clean one) and does not clip around the clean samples either, which allows much larger magnitude of the noise during the attack steps. Doing so can effectively avoid catastrophic overfitting (CO) in which a model that is adversarially trained with single-step FGSM suddenly becomes vulnerable to multi-step attacks. Also, the clean accuracy does not hurt. Experimentally, the proposed N-FGSM method reaches on-par or surpasses the SOTA method GradAlign while enjoying 3x speed-up.


**Questions:**

I do not have questions at this point.


**Limitations:**

Yes.

**Strengths And Weaknesses:**

Strengths:
Preventing the catastrophic overfitting (CO) in which a model suddenly becomes vulnerable to multi-step attacks despite remaining robust to single-step attacks is quite important in adversarial defense. The authors have proposed an adversarial training procedure where the noise is used as a form of data augmentation. The paper has good organization and presentation. Although the proposed method is not significantly different from prior work in terms of mathematical formulation, I do believe that it has its own merit and the authors have provided pretty thorough discussion and analysis in Section 4 and 5, helping readers to understand better the difference between the proposed method and prior work on single-step adversarial training that also incorporates noise. Experimentally, the proposed method performs competitively against prior work. Also, some important ablative studies are also conducted.

Weaknesses:
I think overall this is a solid paper. At this reviewing stage, I do not spot apparent weaknesses.

---

> ### Author Response · Authors · 2022-08-02
> **Author response**
>
> We thank the reviewer for the thoughtful review. We appreciate that the reviewer found that **“The paper has good organization and presentation”** and that our method **“has its own merit”** providing a  **“thorough discussion and analysis [...]  to understand better the difference between the proposed method and prior work”**. We also value that the reviewer found our objective of preventing CO **“quite important in adversarial defense”.**

---

### Official Review · Reviewer_1zew · 2022-07-11

**Rating:** 5
**Confidence:** 4
**Soundness:** 3 good
**Presentation:** 3 good
**Contribution:** 2 fair

**Summary:**

This paper revisits the generation process of adversarial perturbations in adversarial training with single-step FGSM attack. To solve the catastrophic overfitting problem, they propose to add random noises on benign images, and utilize FGSM to optimize adversarial perturbations based on these augment samples without restricting perturbations to lie inside ε-ball. This unrestricted method helps to balance the effect of noises and perturbations. Their experiments demonstrate that the proposed method achieves comparable robustness while being faster than previous methods.

**Questions:**

1. How will the performance of this method change after improving ε of PGD?
2. Why noise augmentation can works as a regularizer?


**Limitations:**

Yes


**Strengths And Weaknesses:**

Strengths
1. The presentation of this paper is good.
2. Fig 1 (left) provides a good illustration for Noise-FGSM. And the analysis between the noise magnitude and the attack strength is interesting.
3. This paper provides a different view to understanding single-step FGSM in adversarial training.

Weaknesses
1. The trade-off between noise magnitude and attack strength should be discussed in detail. Maybe authors can dynamically adjust these two to further prove their points.
2. The clipping ensures that generated adversarial examples are still lie inside the class space. The unrestricted method with a larger noise may undermine this assumption. In addition, the toy experiment (Fig 2 (right)) shows that the unrestricted method works well when the magnitude of noise is equal to 2ε or 4ε. It suggests that larger noise (4ε) may not further prevent CO. The statement of “Larger noise is also necessary to prevent CO” is not rigorous.
3. As shown in line 173-183, N-FGSM obtains a non-distorted loss surface similar to that obtained by GradAlign regularizer. It would be better if the authors could explain why noise augmentation can works as a regularizer?
4. RS-FGSM and GradAlign both performed experiments on ImageNet. Thus, the experiments on ImageNet should also be included in this paper.
5. How will the performance of this method change after improving ε of PGD?
6. Overall, this paper proposes an effective but simple method, and some points are not clear.

---

> ### Author Response · Authors · 2022-08-02
> **Author response 1/2**
>
> We thank the reviewer for the review. We appreciate that the reviewer enjoyed the presentation and found the analysis about noise magnitude and attack strength interesting.
>
> 1.***“The trade-off between noise magnitude and attack strength should be discussed in detail”***.  In Fig 2 (right), we compare the performance of NFGSM as we increase $\epsilon$ (with the FGSM step size alpha fixed to $\epsilon$). Moreover, in Fig 6 (A), we ablate the performance of NFGSM as we vary the FGSM step size alpha and noise magnitude $k$ for a fixed $\epsilon$. We consider this is an exhaustive ablation on those two components. We are not quite sure as if this is what the reviewer meant by “dynamically adjust”?
> The main conclusions based on these experiments are:
>     - To avoid CO we need to increase the noise magnitude (without clipping) compared to previous method RS-FGSM (to least k=2$\epsilon$). See  Fig 2 (right)
>     - In particular, larger $\epsilon$ radii will need larger noise magnitude. See Fig 2 (right)
>     - Increasing the noise magnitude will have a much milder effect on the clean accuracy than increasing the FGSM step size. See Fig 6 (B)
>     - Increasing the FGSM step size (for a fixed $\epsilon$) might require increasing the noise magnitude to avoid CO. See Fig 6 (A).
>
> We look forward to the reviewer’s response on this in case we missed some important ablation or if something is not clear in the text.
>
> 2.1 ***“The unrestricted method with a larger noise may undermine this assumption (i.e. lie outside the $\epsilon$ ball)*** Indeed, it is true that without clipping and noise augmentations the perturbations may lie outside the $\epsilon$ ball. However, **all methods are evaluated against the same test-time attacks**. As long as the evaluation is the same for all methods, it should not matter what training procedure was used during training and whether larger perturbations were used. The reason why training constraints are similar to the test time constraints is to hope for better test performance. But if these constraints during training can be relaxed for better test performance, there is no reason why one should not do that. Additionally, we also show that none of the other methods benefit from larger training $\epsilon$ in preventing CO (see the last paragraph in Section 6).
>
> 2.2 ***“The statement of 'Larger noise is also necessary to prevent CO' is not rigorous.”***
> What we meant is that simply removing clipping is not enough, we also need to increase the noise magnitude compared to what was used in previous methods (e.g. 2$\epsilon$ vs 1$\epsilon$ used in RS-FGSM) to avoid CO (we did not necessarily imply increasing the noise unboundedly). We will clarify this in the text. For further discussion: In Fig 2 (right) we ablate up to 4$\epsilon$ because overly large noise augmentations will also end up in a decrease in performance, but we generally do not expect that stronger noise augmentations would not be able to prevent CO. One should find the sweet spot in terms of noise augmentations: not enough noise will not prevent CO and as we keep increasing the noise, the performance will slowly but surely decrease as well as we start “over-regularizing”.
>
> 3.***“Why noise augmentation can work as a regularizer?”*** We would like to point out that it is not novel to connect noise augmentations with a regularization effect e.g. [1]. However, we fully agree that investigating this further in the context of CO and AT is indeed an interesting direction for future work.
>
> [1] C. M. Bishop: Training with Noise is Equivalent to Tikhonov Regularization. In: Neural Computation (1995)
>
> 4.***Experiments on ImageNet***  As advised, we have conducted experiments on ImageNet and compared our NFGSM against FGSM and RS-FGSM. We observe that FGSM suffers CO for $\epsilon$=6/255 while neither RS-FGSM nor NFGSM do. However, NFGSM enjoys better robustness. For instance, at $\epsilon$=6/255 NFGSM obtains PGD50-10 accuracy of 17.12% while RS-FGSM yields 16.49% and FGSM 0.08% (due to CO). Thus, **NFGSM also avoids CO in ImageNet. We have added these experiments in our submission in Appendix I.** We would like to raise to the attention of the reviewer that the computational cost of the experiments in our paper amount to roughly 2500 GPU hours, which we hope demonstrates the thoroughness and extensiveness of our submission.
>
> 5.***“How will the performance of this method change after improving ε of PGD?”*** We do not think that the test time $\epsilon$ of PGD should be increased for NFGSM compared to other methods. As discussed in 2.1 what matters is that **all methods are evaluated against the same test time attack. Otherwise, this results in an unfair comparison where we are evaluating our NFGSM against stronger attacks than we use to test other methods.**
>
> We continue our response in the next comment.

---

> > ### Author Response · Authors · 2022-08-02
> > **Author response 2/2**
> >
> > 6.***“Overall, this paper proposes an effective but simple method, and some points are not clear.”*** Thank you for the constructive feedback. Hopefully, they are more clear now, please let us know if there are things that remain unclear.
> >
> > As a last remark, we would like to point out that the individual scores for Soundness, Presentation and Contribution are “Good”, “Good” and “Fair”, which we appreciate. We humbly ask the reviewer to reconsider the overall rating taking into account the individual scores.

---

> > > ### Comment · Reviewer_1zew · 2022-08-08
> > > **Reviewer 1zew Response**
> > >
> > > Thanks for the reply. My concerns are well addressed in the rebuttal.

---

> > > > ### Author Response · Authors · 2022-08-08
> > > > **Thank you for the comments and suggestions.**
> > > >
> > > > Thank you for the comments and suggestions, they have definitely helped improve the paper.

---

### Official Review · Reviewer_QcoK · 2022-07-11

**Rating:** 5
**Confidence:** 5
**Soundness:** 2 fair
**Presentation:** 2 fair
**Contribution:** 2 fair

**Summary:**

This paper finds that using a stronger noise around the clean images combined with not clipping can prevent catastrophic overftting and propose Noise-FGSM (N-FGSM) to improve model robustness. Evaluations on some datasets demonstrate that the proposed method outperform existing methods.

**Questions:**

1. supplementary material is same as the submitted manuscript

2. The main problem is that the authors have not compared their approach properly with some other methods that exist in the literature [1][2][3]

3. Lack of comparative experiments on large datasets, such as ImageNet. Previous works all conduct experiments on ImageNet

[1] Andriushchenko, M., Flammarion, N.: Understanding and improving fast adversarial training. In: Annual Conference on Neural Information Processing Systems (2020)

[2] Sriramanan, G., Addepalli, S., Baburaj, A., et al.: Guided adversarial attack for evaluating and enhancing adversarial defenses. Advances in Neural Information Processing Systems 33, 20297–20308 (2020)

[3] Sriramanan, G., Addepalli, S., Baburaj, A., et al.: Towards efficient and effective adversarial training. Advances in Neural Information Processing Systems 34 (2021)

**Limitations:**

Yes

**Strengths And Weaknesses:**

The novelty in my opinion is limited.  Using a larger noise combined with not clipping to conduct fast adversarial training is unfair to other  fast adversarial training methods. The authors should compare the proposed method with other fast adversarial training with  a larger noise combined with not clipping. Moreover, the training epoch is very small. Some research [1][2][3] have found that when the epoch of  fast adversarial training is set to larger, the fast adversarial training can also meet catastrophic overftting, such as 100, 200.

[1] Andriushchenko, M., Flammarion, N.: Understanding and improving fast adversarial training. In: Annual Conference on Neural Information Processing Systems (2020)

[2] Sriramanan, G., Addepalli, S., Baburaj, A., et al.: Guided adversarial attack for evaluating and enhancing adversarial defenses. Advances in Neural Information Processing Systems 33, 20297–20308 (2020)

[3] Sriramanan, G., Addepalli, S., Baburaj, A., et al.: Towards efficient and effective adversarial training. Advances in Neural Information Processing Systems 34 (2021)

---

> ### Author Response · Authors · 2022-08-02
> **Author response**
>
> Thank you for the review,
>
> 1. ***“Using a larger noise combined with not clipping to conduct fast adversarial training is unfair to other fast adversarial training methods”*** We respectfully but strongly disagree with the reviewer for two reasons:
>
>       1. **All methods are evaluated against the *same test-time attack.*** As long as the evaluation is the same for all the methods (fixing other ingredients such as training data etc.), it is absolutely not an unfair comparison. If we follow the same logic, one might say using “random step” is unfair to say FGSM when compared with RS-FGSM. We respectfully ask the reviewer to not consider the simplicity and the obviousness of our approach as its weakness, rather it should be considered as its strength. Moreover, to reconsider the contribution of this work as i) no prior art had thoroughly shown that noise augmentations could be used to prevent CO in all tested settings and ii) having an efficient and effective AT method is of high practical relevance to the robustness community. Therefore, we humbly ask the reviewer to reevaluate their opinion about our work.
>
>       2. Anyhow, we had already conducted various experiments showing that training other methods with larger training $\epsilon$ does not benefit CO (see the last paragraph in Section 6).
>
> 2. ***“Authors should compare the proposed method with other fast adversarial training with a larger noise combined with not clipping.”*** The main finding of this paper is that removing clipping and increasing noise leads to improved performance. This is precisely what we want to ablate so **we cannot apply it to other methods like FGSM or RS-FGSM, otherwise we would end up with our method.** We could apply additional regularizers such as GradAlign to NFGSM but this would defeat the purpose of finding an efficient single-step method since the regularizer is what makes GradAlign expensive.
>
> 3. ***“The training epoch is very small”.*** We focus on efficient adversarial training, thus we employ the same training settings as GradAlign (your reference [1]). However, we also show that **NFGSM can prevent CO even when using longer training schedules as well** (as the reviewer mentioned, 200 epochs). Please refer to Fig 6. (D) and Appendix D.
>
> 4. ***“Supplementary material is same as the submitted manuscript”.*** The supplementary material contains the full paper with Appendix at the end.
>
> 5. ***“The main problem is that the authors have not compared their approach properly with some other methods that exist in the literature [1][2][3]”.*** Actually, **we do compare with [1] under** ***GradAlign.*** Regarding [2] and [3] we had not considered them in our initial comparison because they do not focus on CO and only evaluate for $\epsilon$ 8/255. As suggested, we have now tested the performance of GAT [2] and also NuAT [3] in our settings. We find that both NuAT and GAT suffer from CO for larger $\epsilon$ (with their default settings). Interestingly, if we apply GAT or NuAT regularizers to NFGSM, then we do not observe CO and in most cases a boost in performance occurs. For instance, at $\epsilon$=10/255, GAT has a robust acc (with PGD50-10) of 43.34 +- 0.23 while NFGSM+GAT regularizer obtains 44.97 +- 0.07, in comparison plain NFGSM has 41.56 +- 0.16. This is a compelling result, as it suggests **NFGSM can be combined with other regularizers designed to improve FGSM performance (e.g. GAT or NuAT) and mutually benefit each other. We have added these new experiments in our submission, Appendix L.**
>
> 6. ***“Lack of comparative experiments on large datasets, such as ImageNet”***. As advised, we have conducted experiments on ImageNet and compared our NFGSM against FGSM and RS-FGSM. We observe that FGSM suffers CO for $\epsilon$=6/255 while neither RS-FGSM nor NFGSM do. However, NFGSM enjoys better robustness. For instance, at $\epsilon$=6/255 NFGSM obtains PGD50-10 accuracy of 17.12% while RS-FGSM yields 16.49% and FGSM 0.08% (due to CO). Thus, **NFGSM also avoids CO in ImageNet. We have added these experiments in our submission in Appendix I.**
>
> We would like to raise to the attention of the reviewer that the computational cost of the experiments in our paper amount to roughly 2500 GPU hours, which we hope demonstrates the thoroughness and extensiveness of our submission.

---

> > ### Comment · Reviewer_QcoK · 2022-08-05
> > **Reviewer QcoK  response**
> >
> > 1. “The training epoch is very small”. In [1][2][3], they have claimed that using early stopping (small epoch）can delay CO, but it does not prevent CO. Using larger epcoh can also meet CO. And some new works[4][5] also support this view. it is not clear whether the proposed method can prevent CO. It may be because using early stopping (small epoch）can delay CO.
> >
> > [1]Kim H, Lee W, Lee J. Understanding catastrophic overfitting in single-step adversarial training[C]//Proceedings of the AAAI Conference on Artificial Intelligence. 2021, 35(9): 8119-8127.
> >
> >   [2] Sriramanan, G., Addepalli, S., Baburaj, A., et al.: Guided adversarial attack for evaluating and enhancing adversarial defenses. Advances in Neural Information Processing Systems 33, 20297–20308 (2020)
> >
> >   [3] Sriramanan, G., Addepalli, S., Baburaj, A., et al.: Towards efficient and effective adversarial training. Advances in Neural Information Processing Systems 34 (2021)
> >
> >   [4] Jia X, Zhang Y, Wu B, et al. Boosting fast adversarial training with learnable adversarial initialization[J]. IEEE Transactions on Image Processing, 2022.
> >
> >  [5]Jia X, Zhang Y, Wei X, et al. Prior-Guided Adversarial Initialization for Fast Adversarial Training[J]. arXiv preprint arXiv:2207.08859, 2022.
> >
> > 2. “Using a larger noise combined with not clipping to conduct fast adversarial training is unfair to other fast adversarial training methods”
> > Using a lager epsilon to improve the adversarial performance for adversarial training has been proposed in [5](see Table 3). This paper just uses the larger epsilon (without clip) for fast adversarial training. So the novelty of the proposed method in my opinion is limited.
> >
> >   [5] Uncovering the Limits of Adversarial Training against Norm-Bounded Adversarial Examples
> >
> > 3. “GAT has a robust acc (with PGD50-10) of 43.34 +- 0.23 while NFGSM+GAT regularizer obtains 44.97 +- 0.07,” moreover how many trianing epoch is for this result?
> >
> > 4. Lemma N.1 does not prove the validity of the proposed method. In FGSM-RS, the bound of the FGSM-RS is smaller than FGSM which verifies its effectiveness. But the bound of the proposed is larger than them, why does it work?

---

> > > ### Author Response · Authors · 2022-08-05
> > > **Response from the authors**
> > >
> > > 1. **Longer training** -  We already have experiments to show that N-FGSM can prevent CO even with 200 epochs of training, which is a very long training time. See Figure 6 and Section 7 (paragraph 2) and Appendix D.
> > > 2. **Paper released last month** - The referenced paper [5a] was put on arxiv in July, 2022 which is just last month and  well after the submission deadline.
> > > 3. **Larger training perturbation** - [5b] deals with a very different setting as follows:
> > >    - They do not work with single-step methods.
> > >    - All their experiments are about PGD or TRADES adversarial training while our paper is entirely about behavior that is unique to single-step methods.
> > >    - They do not deal with catastrophic overfitting and do not claim either that their method helps in catastrophic overfitting.
> > > Moreover, we show that other tested methods do not benefit from an increased epsilon during training. See last paragraph in section 6.
> > > 4. **GAT and NuAT** - We used the same training length for GAT/NuAT and GAT/NuAT+NFGSM: of 30 epochs to follow the main setting in our paper. We see that GAT/NuAT suffer CO for larger epsilons whereas combining them with NFGSM does not.
> > > 5. **Lemma N.1** - Indeed, lemma N.1 does not prove the validity of our method and we never claim that it does. As discussed in the caption of Figure 14, the purpose of the lemma is to highlight that, the claim in [6]  about the noise being helpful due to reducing the l2 norm is not correct as, in our case, the noise increases the l2 norm and is still helpful.
> > >
> > >
> > >
> > > [5a] Jia X, Zhang Y, Wei X, et al. Prior-Guided Adversarial Initialization for Fast Adversarial Training[J]. arXiv preprint arXiv:2207.08859, 2022
> > > [5b] Uncovering the Limits of Adversarial Training against Norm-Bounded Adversarial Examples.
> > > [6] Maksym Andriushchenko and Nicolas Flammarion. Understanding and improving fast adver342 sarial training. In Neural Information Processing Systems (NeurIPS), 2020.

---

> > > > ### Comment · Reviewer_QcoK · 2022-08-05
> > > > **Response from the reviewer QcoK**
> > > >
> > > > 1. What worries me most is that the performance of the proposed method is not as good as the previous state-of-the-art work(under AA,  GAT and NuAT achieve about 49%) ? Although the authors show that under larger perturbations, their method is better than other methods (not using the original training setting of compared method), and adversarial training methods are currently commonly conducted under the epsilon=8/255.
> > > >
> > > > 2. Another key issue is that using larger perturbations to improve performance has been proposed in multiple-step adversarial training, and the proposed method only uses this technique in the single-step adversarial training.
> > > >
> > > > 3. From Table 2,  under the long schedule, the performance of the final model is much worse than the best performance of the best model. It may be the proposed method meets CO but overcomes it in late period. I would like to see changes in robustness throughout the whole training.

---

> > > > > ### Author Response · Authors · 2022-08-05
> > > > > **Response from the authors**
> > > > >
> > > > > 1- **Not just increasing perturbation** We would highlight that the main contribution of our work is not merely **using larger epsilon**. The main contribution of our work is providing the **first** *simple* and *efficient* algorithm that avoids the phenomenon of catastrophic overfitting thoroughly.
> > > > >
> > > > >    More specifically addressing the concern of the reviewer, in section 6 of our work, we mention “We observe in Figure 6 (A) that training without noise (essentially, FGSM) leads to CO, with robust accuracy equal to zero, even for large values of the FGSM step $\alpha$. This indicates that it is not an increase in the perturbation norm, but the combination with noise which plays an essential role in circumventing CO for N-FGSM.” The previous work mentioned by the reviewer does not involve adding noise.
> > > > >
> > > > > 2- **Long training schedule plots** In the Appendix D in Figure 10, we show the whole training of N-FGSM and FGSM. Clearly, FGSM suffers CO whereas N-FGSM does not even for long training schedules. Note that, the **gradually decreasing** PGD accuracy of N-FGSM is an example of *robust overfitting*. This was already described in previous work [2] and discussed by us in the text (Lines 309-313). We hope this helps ease the reviewer’s doubt about whether N-FGSM encounters CO and then overcomes it.
> > > > >
> > > > > 3- **Large epsilons for GAT/NuAT** As argued in works like GradAlign [1], being robust must also include experiments with larger epsilons than commonly used epsilons like 8/255. GAT and NuAT do not explore larger epsilons in their work because they do not focus on the phenomenon of CO. As our primary focus is on CO, we explore larger epsilons (epsilon=16/255) following the guidelines of [1] and show that GAT and NuAT suffer from CO whereas GAT/NuAT+NFGSM do not.
> > > > >
> > > > >    We want to stress that we are not proposing NFGSM as an alternative to methods like GAT/NuAT but rather that these methods can be efficiently combined with N-FGSM which results in further increase of robust performance as well as avoids CO. We will highlight this in the main text.
> > > > >
> > > > > [1] Maksym Andriushchenko and Nicolas Flammarion. Understanding and improving fast adversarial training. In: Annual Conference on Neural Information Processing Systems (2020)
> > > > > [2] Leslie Rice, Eric Wong, and Zico Kolter. Overfitting in adversarially robust deep learning. In International Conference on Machine Learning (ICML), 2020.

---

> > > > > ### Comment · Reviewer_QcoK · 2022-08-05
> > > > > **Response from the reviewer QcoK**
> > > > >
> > > > > 1. What worries me most is that the performance of the proposed method is not as good as the previous state-of-the-art work(under AA, GAT and NuAT achieve about 49%) ?
> > > > >
> > > > > 2. Why does the proposed method can prevent CO？
> > > > >
> > > > > 3. "We observe in Figure 6 (A) that training without noise (essentially, FGSM) leads to CO, with robust accuracy equal to zero, even for large values of the FGSM step " .This phenomenon has been discovered before. The paper is not the first one.
> > > > >
> > > > > 4. The paper claim 'Augment sample with additive noise', it is same with FGSM-RS( starting with the uniform noise). Why introduce a new concept?

---

> > > > > > ### Author Response · Authors · 2022-08-05
> > > > > > **Author response**
> > > > > >
> > > > > > 1- **"GAT/NuAT performance"** Indeed the performance obtained with the short training schedule is lower than in their work since we used the short training schedule. The main point of this work is to obtain a **fast** adversarial training method while avoiding CO and we do observe that GAT and NuAT suffer CO for larger epsilon. Therefore, *although they do have impressive performance at 8/255, they are not the main baseline to look at.* **The only prior work that has shown could avoid CO in all settings tested in this paper is GradAlign.** Moreover, we have seen N-FGSM can be succesfully combined with them therefore **they are not competing methods, prectitioners can just use them together** if they are willing to spend the extra compute required for GAT and NuAT regularizers.
> > > > > >
> > > > > > 2- **"Why does N-FGSM prevent CO?"** The FGSM attack relies on the loss being locally linear for the approximation of the worst case adversary to be correct (as also pointed out in [1] and [2]). By analyzing the loss landscape, we observe that both FGSM and RS-FGSM lead to a highly non-linear surface (which renders FGSM attacks ineffective). On the other hand, both our proposed **N-FGSM and GradAlign lead to a locally linear loss which leads to an effective FGSM attack throughout training preventing CO (see Fig. 13 in Appendix).**
> > > > > >
> > > > > > GradAlign explicitly incorporates a regularizer to enforce the loss surface to be locally linear. However, N-FGSM does not. Therefore, based on the fact that noise do have regularization impact that encourages Lipschitzness [3], and the observed similarity between the N-FGSM and GradAlign loss surfaces, we hypothesize that the effectiveness of N-FGSM is because of the implicit regularization of the noise that enforces the loss to be locally linear.
> > > > > >
> > > > > > 3- **"Figure 6A"** If we could ask the reviewer to revisit Fig 6A, it does not simply show that FGSM with larger alpha does not succed at preventing CO but that when increasing the noise augmentations rather than alpha one can avoid it. Moreover, it also shows that (without clipping) stronger noise augmentations (e.g. 2epsilon) can prevent CO for larger values of alpha which were showed not to work with settings of RS-FGSM [4]. All in all, it shows that *N-FGSM can prevent CO thanks to the noise augmentations rather than simply increasing the perturbations as done in previous work mentioned by the reviewer.* **To the best of our knowledge the findings in Figure 6 have not been shown in any prior work, please do provide a reference if that is inacurate.**
> > > > > >
> > > > > > 4- **"Noise data augmentations"** Our motivation for removing clipping and increasing the magnitude of noise perturbations comes from the intuition of considering noise as data augmentation rather than a random initialization as part of the attack. For a more detailed motivation and discussion please refer to Sec 4 of our paper.
> > > > > >
> > > > > > [1] M. Andriushchenko, N. Flammarion: Understanding and improving fast adversarial training. In: Annual Conference on Neural Information Processing Systems (2020)
> > > > > > [2] H. Kim, W. Lee., J. Lee: Understanding Catastrophic Overfitting in Single-step Adversarial Training. In: Association for the Advancement of Artificial Intelligence (2021)
> > > > > > [3] C. M. Bishop: Training with Noise is Equivalent to Tikhonov Regularization. In: Neural Computation (1995)
> > > > > > [4] Wong et. al.: Fast is better than free: Revisiting adversarial training. In: ICLR 2020

---

> > > > > > > ### Comment · Reviewer_QcoK · 2022-08-08
> > > > > > > **Reviewer response**
> > > > > > >
> > > > > > > Thanks for the reply. My concerns are well addressed in the rebuttal.

---

> > > > > > > > ### Author Response · Authors · 2022-08-08
> > > > > > > > **Thank you for the comments and suggestions.**
> > > > > > > >
> > > > > > > > Thank you for the comments and the discussion throughout the rebuttal. Your suggestions have definitely helped improve the paper.

---

### Official Review · Reviewer_khDd · 2022-07-12

**Rating:** 5
**Confidence:** 5
**Soundness:** 3 good
**Presentation:** 3 good
**Contribution:** 3 good

**Summary:**

This paper focus on the catastrophic overfitting issue in FGSM adversarial training. The work proposes to avoid clipping adversarial perturbations to mitigate the CO and they regard adding noise as a data augmentation method. This paper analyze the proposed tricks in detail and provide a huge of experimental results to prove them.

**Questions:**

- Do you provide the comparisons with other single-step methods against AutoAttack?
- Do you consider to compare with this single-step AT method[1]?

Citations:
[1] Guided Adversarial Attack for Evaluating and Enhancing Adversarial Defenses

**Limitations:**

See above

**Strengths And Weaknesses:**

Strengths:
- This work is easy to follow.  And authors use clear math formulas to show their methods. Their experimental results look good.
- They regard the random initialization  as a data augmentation trick which can help explain many problems.

Weakness:
- Although the authors provide some theoretical proofs and the loss landscapes,  the proof and figures are more about the influence brought by the noise augmentation. They dose not show why the larger noise magnitude can prevent the CO better.

---

> ### Author Response · Authors · 2022-08-02
> **Author response**
>
> We thank the reviewer for their constructive feedback. We appreciate that you found our work “*easy to follow*” and our “*experimental results good*”.
>
> \- ***How noise augmentations can prevent CO better.*** The FGSM attack relies on the loss being locally linear for the approximation of the worst case adversary to be correct (as also pointed out in [1] and [2]). By analyzing the loss landscape, we observe that both FGSM and RS-FGSM lead to a highly non-linear surface (which renders FGSM attacks ineffective). On the other hand, both our proposed **N-FGSM and GradAlign lead to a locally linear loss which leads to an effective FGSM attack throughout training preventing CO (see Fig. 13 in Appendix).**
>
>  GradAlign explicitly incorporates a regularizer to enforce the loss surface to be locally linear. However, N-FGSM does not. Therefore, based on the fact that noise do have regularization impact that encourages Lipschitzness [3], and the observed similarity between the N-FGSM and GradAlign loss surfaces, we hypothesize that the effectiveness of N-FGSM is because of the implicit regularization of the noise that enforces the loss to be locally linear.
>
> We agree that rigurously showing how are the noise augmentations regularizing the loss surface in the context of adversarial training would be very interesting. However, this analysis is a challenging problem in itself and we leave it as future work.
>
>
> \- ***Testing with AutoAttack.*** In Appendix G we present evaluations of AutoAttack comparing NFGSM against FGSM and GradAlign. We observe similar patterns as with PGD50-10 where **NFGSM maintains equally high robustness to GradAlign, this shows that our approach is effective across different strong attacks.**
>
> \- ***Comparing to GAT.*** Following the reviewer’s advise, we have tested the performance of GAT (and also NuAT [4]) in our settings. We find that both NuAT and GAT suffer from CO for larger $\epsilon$ (with their default settings). Interestingly, if we apply GAT or NuAT regularizers to NFGSM then we do not observe CO and usually a boost in performance. For instance, at $\epsilon$=10/255, GAT has a robust acc (against PGD50-10) of 43.34 +- 0.23 while NFGSM+GAT regularizer obtains 44.97 +- 0.07, in comparison plain NFGSM has 41.56 +- 0.16. This is a compelling result, as it suggests **NFGSM can be combined with other regularizers designed to improve FGSM performance (e.g. GAT) and mutually benefit each other. We have added these new experiments in our submission, Appendix L.**
>
> [1] M. Andriushchenko, N. Flammarion: Understanding and improving fast adversarial training. In: Annual Conference on Neural Information Processing Systems (2020)
> [2] H. Kim, W. Lee., J. Lee: Understanding Catastrophic Overfitting in Single-step Adversarial Training. In:  Association for the Advancement of Artificial Intelligence (2021)
> [3] C. M. Bishop: Training with Noise is Equivalent to Tikhonov Regularization. In: Neural Computation (1995)
> [4] Sriramanan, G., Addepalli, S., Baburaj, A., et al.: Towards efficient and effective adversarial training. Advances in Neural Information Processing Systems (2021)

---

> > ### Comment · Reviewer_khDd · 2022-08-07
> > **Reviewer khDd Response**
> >
> > Thanks for your reply.
> > 1. The loss surface can only show that your method can lead to a locally linear loss. Why do your method or random noise can make the loss locally linear?

---

> > > ### Author Response · Authors · 2022-08-07
> > > **Thanks and Author response**
> > >
> > > We thank the reviewer for their response and their questions. Below, we explain our answer with multiple references to literature and we can include this in the final draft too.
> > >
> > > * **Adding noise is equivalent to regularisation** -  In Bishop et. al. [a], the authors show that adding noise to the input during training is equivalent to adding an extra regulariser during training where the regulariser is penalises the local lipschitznes exactly (see Eq 18 in [a] and the discussion before it).
> > > * **Noise sensitivity is equivalent to local lipscitzness** - Arora et. al. [b]  show in Proposition 3.1 that noise sensitivity of a NN layer is exactly equivalent to its local lipschitzness. As a corollary, it implies that regularising noise sensitivity also regularises local lipschitness. The other way (i.e. penalising local lipschitzness decreases noise sensitivity) was shown extensively in Sanyal et. al [c].
> > >
> > > Combining these empirical and theoretical evidences, we believe that the effectiveness of N-FGSM is because of the implicit regularisation of the noise that enforces the loss to be locally linear.
> > >
> > >
> > > [a] C. M. Bishop: Training with Noise is Equivalent to Tikhonov Regularization. In: Neural Computation (1995).
> > > [b] Arora, Sanjeev, et al. "Stronger generalization bounds for deep nets via a compression approach." International Conference on Machine Learning. PMLR, 2018.
> > > [c] Sanyal, Amartya, Philip HS Torr, and Puneet K. Dokania. "Stable rank normalization for improved generalization in neural networks and gans." International Conference on Learning Representations (2020).

---

### Author Response · Authors · 2022-08-05
**Gentle nudge | We request you to let us know if there is anything that still requires clarification**

Dear Reviewers,

We hope that we have satisfactorily replied to all your concerns. Our replies are supported heavily by new experiments and conceptual justifications.

We understand that you must be super occupied so apologies for the reminder. We request you to have a look at our reply and let us know if there is anything that isn’t clear yet. We will be happy to get back to you to clarify any doubts or questions you may still have.
Looking forward to a constructive discussion.

Thanks!

---

### Meta-Review · Area_Chair_zh9h · 2022-08-25

**Recommendation:** Accept
**Confidence:** Certain

**Metareview:**

This paper enhances single-step adversarial training by adopting a much stronger noise for initialization. The initial concerns were mostly about missing ablations and misunderstandings/confusions, which were well addressed in the rebuttal. As a result, all reviewers unanimously agree to accept this submission.

In the final version, the authors should include all the clarifications and the additional empirical results provided in the rebuttal.

**Award:**

No

---

### Decision · Program_Chairs · 2022-09-14

Accept